# Dietary protein increases T-cell-independent sIgA production through changes in gut microbiota-derived extracellular vesicles

Jian Tan[1,2], Duan Ni[1,2], Jemma Taitz[1,2], Gabriela Veronica Pinget[1,2], Mark Read [1,3], Alistair Senior [1,4], Jibran Abdul Wali [1,4], Reem Elnour[1,4], Erin Shanahan[1,4], Huiling Wu[1,5,6], Steven J. Chadban [1,5,6], Ralph Nanan[1,7], Nicholas Jonathan Cole King [1,2,8], Georges Emile Grau [2,9], Stephen J. Simpson [1,4] & Laurence Macia [1,2,8 ✉]

Secretory IgA is a key mucosal component ensuring host-microbiota mutualism. Here we use nutritional geometry modelling in mice fed 10 different macronutrient-defined, isocaloric diets, and identify dietary protein as the major driver of secretory IgA production. Protein-driven secretory IgA induction is not mediated by T-cell-dependent pathways or changes in gut microbiota composition. Instead, the microbiota of high protein fed mice produces significantly higher quantities of extracellular vesicles, compared to those of mice fed high-carbohydrate or high-fat diets. These extracellular vesicles activate Toll-like receptor 4 to increase the epithelial expression of IgA-inducing cytokine, APRIL, B cell chemokine, CCL28, and the IgA transporter, PIGR. We show that succinate, produced in high concentrations by microbiota of high protein fed animals, increases generation of reactive oxygen species by bacteria, which in turn promotes extracellular vesicles production. Here we establish a link between dietary macronutrient composition, gut microbial extracellular vesicles release and host secretory IgA response.

[1] Charles Perkins Centre, The University of Sydney, Sydney, NSW, Australia. [2] School of Medical Sciences, Faculty of Medicine and Health, University of Sydney, Sydney, NSW, Australia. [3] School of Computer Science, Faculty of Engineering, University of Sydney, Sydney, NSW, Australia. [4] School of Life and Environmental Sciences, Faculty of Science, University of Sydney, Sydney, NSW, Australia. [5] Kidney Node Laboratory, The Charles Perkins Centre, University of Sydney, Sydney, NSW, Australia. [6] Renal Medicine, Royal Prince Alfred Hospital, Sydney, NSW, Australia. [7] Sydney Medical School Nepean, University of Sydney, Sydney, NSW, Australia. [8] Sydney Cytometry, The University of Sydney and The Centenary Institute, Sydney, NSW, Australia. [9] Vascular Immunology Unit, The University of Sydney, Sydney, NSW, Australia. ✉email: Laurence.macia@sydney.edu.au

The gut microbiota, defined by the trillions of bacteria that inhabit the gut, play a critical role in the physiology and immunity of the host[1]. To maintain a symbiotic relationship, strategies have been developed by the host to regulate its interaction with the gut microbiota. Mucosal secretory IgA (sIgA) plays a key role in this host-microbiota mutualism by excluding pathobionts and pathogens and limiting bacterial attachment to the epithelium[2]. While patients with selective IgA deficiency appear healthy, they are more susceptible to various diseases, including gastrointestinal disorders and allergies[3], highlighting the importance of IgA in mucosal homoeostasis.

Under homoeostatic conditions, IgA is primarily produced by plasma cells local to the small intestine lamina propria through a T cell-independent pathway[4,5]. The majority of commensal bacteria induce T-cell-independent IgA responses[6], and sIgA generated via this pathway is of low affinity and polyreactive towards common bacterial antigens, limiting the attachment of a diverse range of bacteria to the host epithelium[7]. The induction of sIgA is regulated by the local cytokine environment, with the increase in class switch recombination (CSR) cytokines, APRIL and BAFF, produced by epithelial cells, promoting the differentiation of IgM+ B cells into IgA-producing plasma cells[5,8]. These cytokines are produced as a result of toll-like receptor (TLR) activation, particularly TLR4, which recognises bacterial lipopolysaccharide (LPS)[9].

TLR activation by commensal bacteria is critical both for the recruitment of IgM+ B cells in the lamina propria, by increasing the expression of the gut homing chemokine CCL28[9], and for the induction IgA CSR by upregulating APRIL[10]. TLR4 is the major TLR expressed on the epithelium and its overstimulation in TLR4 transgenic mice has been shown to promote CCL28 and APRIL production and accumulation of IgA in the lamina propria and gut lumen[9]. The majority of IgA produced in the gut is in dimeric form, which can be transported through the epithelium into the lumen by binding the polymeric immunoglobulin receptor (pIgR)[11]. Similarly, pIgR expression can be upregulated by several inflammatory cytokines such as IFN-γ, IL-1, IL-17, TNF, and IL-4, following TLR4/NFκB signalling[12]. Of note, IL-4 can have a dual role by also promoting IgA CSR.

Seminal work utilising germ-free mice has established that the presence of gut bacteria is the main driver of sIgA response, and transient colonisation of germ-free animals with *E. coli* leads to transient sIgA production[13]. However, how dynamic changes to the gut microbiota composition affect sIgA is less clear. Diet composition is a major driver of the microbiota composition with most studies focusing on microbiota profiling in the caecum and the colon[14], where IgA production is minimal, compared to the small intestine. How dietary macronutrient composition (i.e., protein, fat and carbohydrate) affects the dynamic interplay between the small intestine microbiota and mucosal IgA remains unknown. This knowledge could establish which diet compositions are beneficial for restoring gut homoeostasis and which diets are detrimental.

In our approach, we fed mice on one of 10 different isocaloric diets varying in their macronutrient composition, which identified an association between dietary protein and sIgA levels using mixture modelling. We validated and explored these results further using a subset of the 10 diets, representing a high-protein, high-carbohydrate or high-fat diet. We found that the high-protein diet had the highest intestinal sIgA levels, and this was associated with increased expression of CCL28 and APRIL in the small intestine. These changes were correlated with the increased capacity of the microbiota to stimulate TLR4 directly or indirectly via increased production of gut microbiota-derived extracellular vesicles (EV). We also established that EV derived from high-protein diet microbiota could directly promote the expression of CCL28. Our work highlights the key role of dietary protein on

sIgA production and identifies bacterial-derived EV as a mediator of gut microbiota-host mutualism.

## Results

**High protein feeding promotes high lamina propria IgA production and higher secretion of luminal sIgA.** To determine how dietary macronutrients might affect sIgA and thus host-microbiota mutualism, we fed mice on one of 10 isocaloric diets with defined ratios of macronutrients in the ranges 5–60% protein, 20–75% fat, and 20–75% carbohydrate, for at least 6 weeks (Fig. 1a and Supplementary Table 1).

The impact of macronutrient composition on gut luminal concentrations of sIgA was visualised using a proportion-based nutritional geometry approach, as described previously[15,16]. Mixture models were used to determine the effects of diet composition on luminal sIgA concentration, as quantified by ELISA ($n = 8$ per diet). Predicted effects of diet on sIgA were mapped onto a right-angled mixture triangle plot, where protein concentration in the diet is represented on the $x$ axis, fat on the $y$ axis and carbohydrate on the hypotenuse (Fig. 1b). Regions of the nutrient mixture space appearing in red demonstrate high levels of sIgA while areas in deep blue represent low levels of sIgA, and values on the isolines indicate the modelled concentration of sIgA (ng/ml). Of the 4 models fitted to sIgA concentration (relating to the first to fourth-order of the Scheffé polynomials, described by Lawson and Wilden[17]), model 1 was the most appropriate as indicated by the lowest Akaike Information Criterion (AIC) value (Supplementary Table 2), which reveal that dietary protein concentration was the principal predictor of sIgA, there being a clearly graded increase in sIgA with increasing protein concentration, irrespective of fat or carbohydrate content ($R^2 = 0.8076$). Linear regression revealed that this effect was driven significantly by protein, but not by carbohydrate or fat intake, and was independent of total caloric intake (Supplementary Fig. 1a–d). Accordingly, when the three diets located at the apices of the mixture space in Fig. 1a were compared, the highest concentrations of gut luminal sIgA were observed in mice fed on the highest protein diet (HP; P60 C20 F20), compared to those fed on a diet high in fat (HF; P5 C20 F75) or high in carbohydrate (HC; P5 C75 F20) (Fig. 1c).

The shape of the response surface for plasma IgA was markedly different from that of sIgA, with plasma IgA being lowest on diets containing high-protein coupled with both low fat and low carbohydrate (Fig. 1d, e and Supplementary Table 3, $R^2 = 0.7243$). Macronutrient composition was not a predictor of plasma IgM concentrations, as the null model was determined to be most favourable by AIC (Supplementary Table 4). This suggests that high luminal sIgA under high-protein feeding conditions was not linked to a systemic increase in basal B cell activity and that this effect was, therefore, mucosa-specific.

To determine whether elevated sIgA levels in the gut lumen of HP-fed mice were due to higher local production of IgA in the lamina propria, we performed immunofluorescence staining with anti-IgA on frozen sections of small intestine isolated from mice fed on HP, HC, or HP diets. Immunofluorescence analysis showed that HP feeding led to the highest expression of IgA in the lamina propria, compared to HC or HF feeding (Fig. 1f). Consistent with this, mice fed on an HP diet had a significantly greater number of B220−IgA+ plasma cells in the small intestinal lamina propria (Fig. 1g), as determined by flow cytometry (gating strategy presented in Supplementary Fig. 1e). Prior to differentiation into IgA-producing plasma B cells, B cells are recruited to the gut by the gut epithelial chemokine CCL28. By qPCR analysis, we found that mice fed on HP diet had significantly higher intestinal gene expression of the B cell gut homing chemokine, CCL28

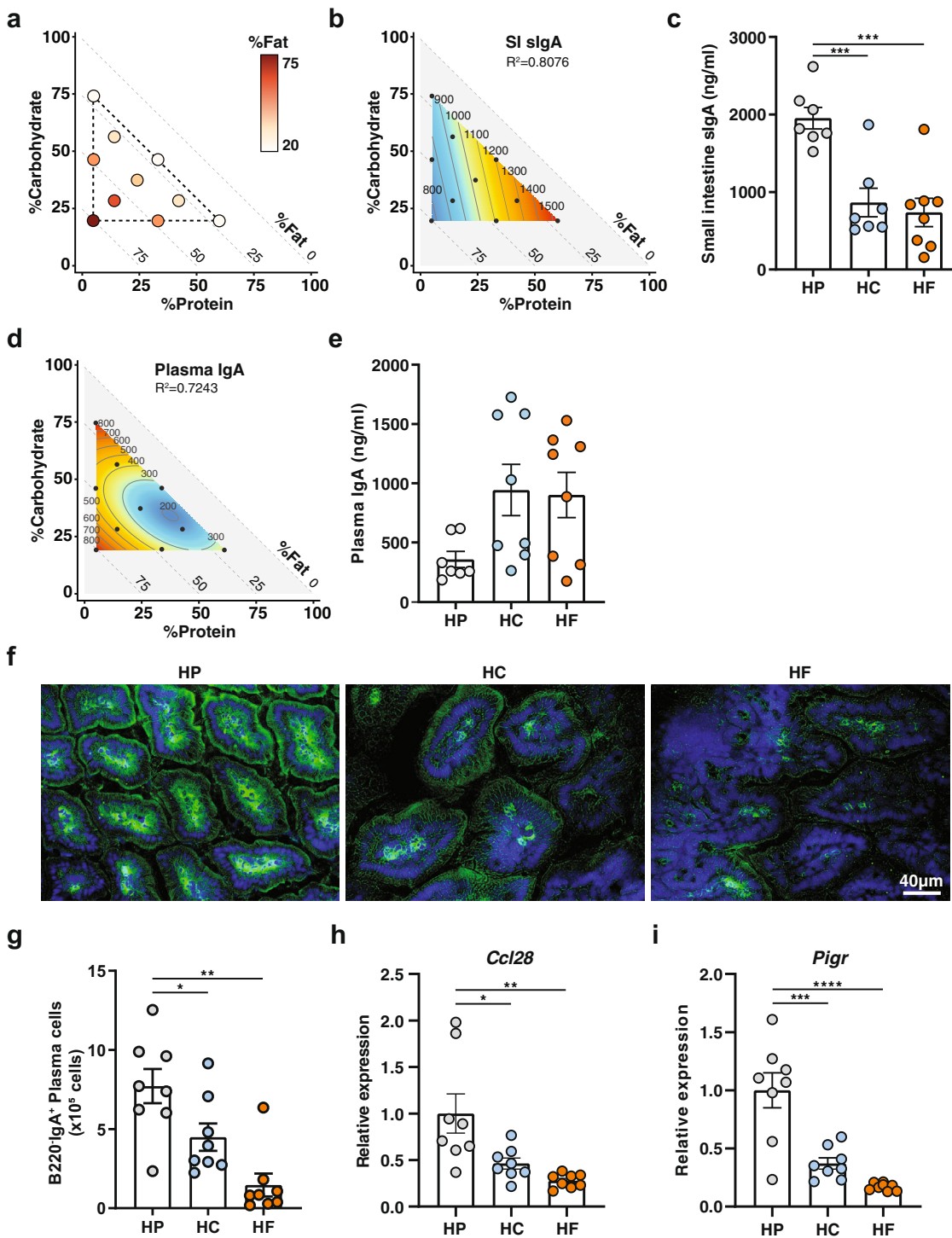

(Fig. 1h). Additionally, the expression of the gene encoding for pIgR, involved in the transport of sIgA to the lumen via the epithelium, was significantly increased under HP feeding conditions (Fig. 1i).

Finally, we observed that a minimum of 5 weeks on HP feeding was necessary to stably increase sIgA levels (Supplementary Fig. 1f) and that this effect was a reversible process, as sIgA levels in HP-fed animals decreased when switched to an HF diet (Supplementary Fig. 1g).

Together, these data show that protein is the major macronutrient driving sIgA production in the gut lumen and that this is reversible. An HP diet promotes the expression of the B cell gut homing chemokine, CCL28, and accordingly, a higher presence of IgA in the small intestine lamina propria. Finally, an HP diet also promotes increased expression of pIgR, the transporter of IgA in the gut lumen, consistent with the highest concentration of luminal sIgA.

**High protein feeding promotes IgA production through T-cell-independent mechanisms.** IgA-producing plasma cells in the lamina propria either originate from IgM-producing B cells that differentiate locally in the lamina propria, or from IgA-expressing plasmablasts induced in gut-associated lymphoid structures such as the Peyer's patches or the mesenteric lymph nodes, which

**Fig. 1 High protein feeding promotes high lamina propria IgA production and higher luminal sIgA.** Animals were fed one of ten isocaloric diets encompassing a macronutrient range of protein (5–60%), carbohydrate (20–75%), and fat (20–75%) for 6 weeks. **a** Visual representation of the composition of the diets used in this study. Each diet is represented by one circle each and their localisation on the x axis and on the y axis define their proportion of protein and of carbohydrate, respectively. The proportion of fat is indicated by the colour range as illustrated in the legend. **b** Contribution of macronutrient composition to small intestinal luminal sIgA ($n = 7$–8 per diet, quantified by ELISA) was modelled by mixture model and represented on a right-angled mixture triangle comprising of carbohydrate (y axis), protein (x axis) and fat (hypotenuse) with small intestinal content IgA concentration (ng/ml, numbers on isolines) as the response variable. Red represents high levels of sIgA while blue represents low levels of sIgA in the nutrient mixture space. Each dot represents one of the 10 diets used for modelling response surface. **c** Scatter bar plot of sIgA from mice fed on a high-protein (HP), high-carbohydrate (HC) or high-fat (HF) diet as determined by ELISA ($n = 7$ or $n = 8$ mice per diet for HP/HC and HF diet respectively; HP vs. HC $p = 0.0009$, HP vs. HF $p = 0.0002$). **d** Mixture model of plasma IgA represented on a right-angled mixture triangle and (**e**) corresponding scatter bar plot ($n = 7$ or $n = 8$ mice per diet for HP, and HC/HF diet respectively). **f** Representative immunofluorescence staining of IgA (green) in the small intestine counterstained with DAPI (blue) from mice fed on an HP, HC, or HF diet for 5 weeks. The scale bar represents 40 μm. **g** Total number of B220$^-$IgA$^+$ IgA plasma cells in the small intestine lamina propria as determined by flow cytometry ($n = 8$ mice per group; HP vs. HC $p = 0.0489$, HP vs. HF $p = 0.0002$). **h, i** Gene expression of (**h**) Ccl28 (HP vs. HC $p = 0.0187$, HP vs. HF $p = 0.0017$) and (**i**) Pigr (HP vs. HC $p = 0.0002$, HP vs. HF $p = <0.0001$) in whole small intestine tissue was determined by qPCR from mice fed on either a HP, HC, or HF diet ($n = 8$ mice per group). Data are represented as mean ± SEM. Results represent $n = 2$ (**e–i**) and $n = 3$ independent experiments (**c**). *$p < 0.05$, **<0.01, ***<0.001, ****<0.0001 by ordinary one-way ANOVA followed by Tukey's multiple comparisons test.

migrates back into the lamina propria to differentiate into mature plasma cells. To identify the origin of the IgA$^+$ plasma cells that are increased under HP feeding conditions, we assessed by flow cytometry the proportion of total B cells, as well as IgA plasmablasts in both Peyer's patches, the main site of T-cell-dependent IgA plasma cell induction, and mesenteric lymph nodes of mice fed on HP, HC or HF diets. We found that proportions of total B cells in both the Peyer's patches (Fig. 2a) and mesenteric lymph nodes (Fig. 2b) were similar between groups. Likewise, no changes in the proportion of IgA$^+$B220$^+$ plasmablast were observed between groups in both the Peyer's patches (Fig. 2c, d) and the mesenteric lymph nodes (Supplementary Fig. 2a). This suggests that HP feeding does not increase T-cell-dependent IgA production, as confirmed by the similar proportions of GL7$^+$CD95$^+$ germinal centre B cells in both the Peyer's patches (Fig. 2e, f) and the mesenteric lymph nodes (Supplementary Fig. 2b). To confirm this, we depleted CD4$^+$ T cells in mice fed on an HP diet from the start of the dietary intervention and found that mice treated with isotype or anti-CD4 depleting antibodies had comparable levels of sIgA (Supplementary Fig. 2c, d). Mice fed on a control diet had lower sIgA than HP-fed mice regardless of the depletion of CD4$^+$ T cells. Together, our results show that HP feeding promotes high levels of small intestine sIgA via a T cell-independent pathway.

**High-protein feeding modulates the lamina propria cytokine environment, specifically APRIL, to favour IgA production.** T-cell-independent induction of IgA CSR results from the complex interaction between gut microbes, host gut epithelial cells, immune cells and the stromal cells of the lamina propria. TLR activation in epithelial cells leads to the production of A Proliferation-inducing Ligand (APRIL) and B cell-activating factor (BAFF), the major cytokines in B cell IgA CSR, as well as thymic stromal lympho-poietin (TSLP). TSLP further amplifies this signal by inducing APRIL and BAFF production by dendritic cells[18].

To determine whether diet composition affects these cytokines, we quantified their expression levels in the small intestines of mice fed on HP, HC and HF diets by qPCR. The expression of April was significantly higher under HP feeding conditions, approximately twofold greater than HC- and HF-fed mice (Fig. 3a). Similarly, Baff was highly expressed under HP feeding conditions, compared to HF feeding conditions but HC-fed mice had levels of Baff expression similar to HP-fed mice (Fig. 3b). This suggests that increased expression of April might account for elevated sIgA levels under HP feeding conditions. Like Baff, Tslp expression was significantly lower in HF-fed mice, whilst HP- and

HC-fed mice had similar higher levels of expression (Fig. 3c). TSLP biases T-cell differentiation towards Th2 T cells which are characterised by their production of the cytokine IL-4. IL-4 has been shown to promote IgA CSR as well as pIgR expression. Indeed, mice fed on an HP diet had significantly elevated expression of Il4, while HF-fed mice had the lowest expression (Fig. 3d). Among other key cytokines involved in IgA CSR induction, IL-10 and TGF-beta produced by epithelial cells or dendritic cells are co-signals necessary for BAFF and APRIL to mediate their effects. We found that the expression of Tgfb was significantly higher in HP-fed mice and Il10 was significantly lower in HF-fed mice compared to HC groups (Fig. 3e, f).

These data demonstrate that HP feeding induces the highest expression of cytokines involved in IgA CSR, and that these key cytokines are lowest under HF feeding.

**Dietary intervention significantly affects the small intestine microbiota composition.** Diet is one of the most influential factors driving gut microbiota composition[19], which in turn modulates host IgA responses. To characterise the impact of HP, HC and HF diet feeding on the small intestinal gut microbiota composition, we performed 16 S rRNA DNA sequencing of small intestine luminal samples. The analysis demonstrated equal sequencing reads between samples (raw data presented in Supplementary Table 5 and corresponding graphs in Supplementary Fig. 3a, b). Furthermore, rarefaction analysis revealed adequate sequencing depth (total number of sequencing reads), with a horizontal asymptote evident for all samples (Supplementary Fig. 3c), demonstrating that our sequencing depth far exceeded what is required to uncover all observations (number of unique taxa, or ASV) present in each sample.

The diversity of the gut microbiota is often used as a marker of a "healthy microbiome". We found that bacterial richness was similar across all diets (Fig. 4a). However, HF-fed animals had lower diversity measures such as evenness and Inverse Simpson index, where Inverse Simpson index was significantly lower when compared to the HP group (Fig. 4a). By principal component analysis (PCA) of Aitchison distance, a compositionally appropriate between-sample distance metric, mice fed on the different diets had significantly distinct gut microbiota composition (Fig. 4b) as determined by PERMANOVA analysis (Supplementary Table 6). Likewise, principal coordinates analysis (PCoA) of UniFrac distances, a between-sample distance metric that considers phylogenetic relatedness, showed similar results, with HF feeding having the most distinct bacterial communities in

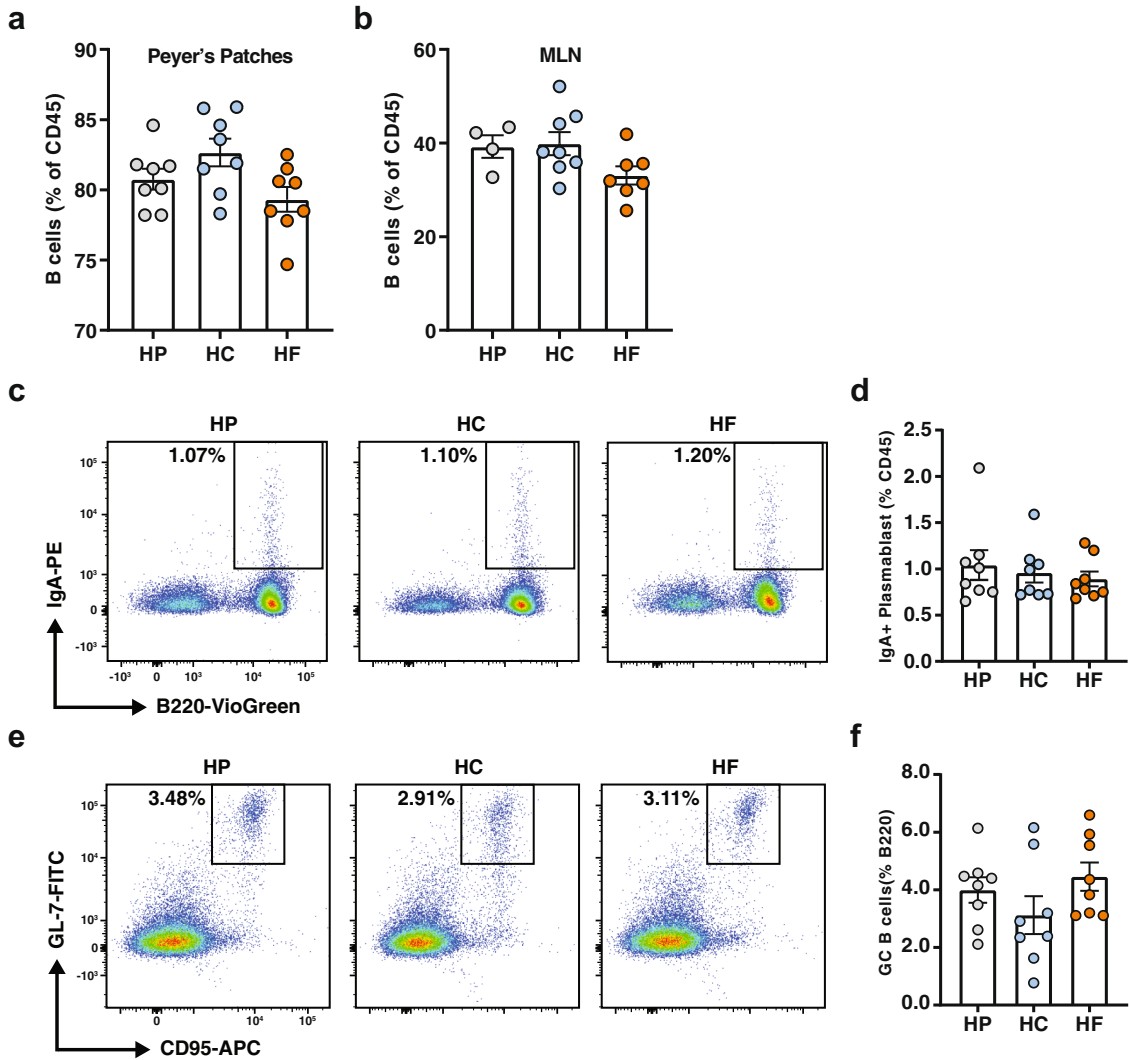

**Fig. 2 High protein feeding promotes IgA production through T-cell-independent mechanisms.** Mice were fed on either a high-protein (HP), high-carbohydrate (HC) or high-fat (HF) diet for 6 weeks. **a** Proportion of B220+ B cells in the Peyer's patches (n = 8 mice per diet) and in the (**b**) mesenteric lymph nodes (MLN; n = 4, n = 8, and n = 7 mice per diet for HP, HC and diet, respectively) were determined by flow cytometry. **c** Representative flow cytometry plots representing the proportion of B220+IgA+ plasmablasts in the Peyer's patches and (**d**) corresponding scatter bar graph (n = 8 mice per diet). **e** Representative flow cytometry plots and graph representing the proportions of CD95+GL7+ germinal centre B cells in the Peyer's patches and (**f**) corresponding scatter bar graph (n = 8 mice per diet). Data are represented as mean ± SEM. Results represent n = 2 independent experiments.

both unweighted and weighted UniFrac PCoA analyses (Supplemental Fig. 3d, e and Supplementary Table 6).

At the phylum level, HF feeding was associated with a higher ratio of *Firmicutes:Bacteroidetes*, as well as increased representation of *Verrucomicrobia*, compared to the microbiota of HP- and HC-fed mice (Fig. 4c). On the other hand, Proteobacteria was underrepresented in the microbiome of HP-fed mice (Fig. 4c). To investigate this further, we applied the ALDEx2 statistical test at the genus level (Fig. 4d) to identify differentially abundant genera (Supplementary Tables 7-9). Consistent with the over-representation of bacteria from the phylum Verrucomicrobia in HF-fed mice (Fig. 4c), *Akkermansia*, which can utilise host mucus as a carbon source[20], was significantly higher compared to both HP- and HC- fed mice (Fig. 4e). The microbiota of the HP-fed mice was characterised by the increased abundance of bacteria from the phylum *Actinobacteria* (Fig. 4c) with the over-representation of bacteria from the genus *Bifidobacterium* (Fig. 4f). Finally, HC feeding was characterised by the over-representation of bacteria from the genus *Allobaculum* (Fig. 4g). Of note, we did not identify segmented filamentous bacteria or

*Mucispirillum* in our animals (Supplementary Tables 7-9), which are two genera known to induce strong T-cell-dependent sIgA responses[6].

Overall, macronutrient composition significantly impacts both the alpha and beta diversity of the small intestine microbiome, with fat appearing to be the dominant driver.

**EV derived from high-protein-fed microbiota activate epithelial TLR4 and promote the expression of PIGR and APRIL.** While we identified that each diet had an impact on the gut microbiota, we then investigated the mechanisms through which the HP microbiota promote the host sIgA response. TLR4 is one of the main TLRs expressed by the small intestinal enterocytes and TLR4 signalling is the major pathway promoting T-cell-independent IgA responses[9]. To determine whether TLR4 signalling was linked to the effect of HP in vivo, we fed wild type (WT) versus *Tlr4⁻/⁻* mice on HP diets for 6 weeks and measured sIgA levels in the small intestine. The absence of TLR4 abrogated the effects of HP on sIgA levels (Supplementary

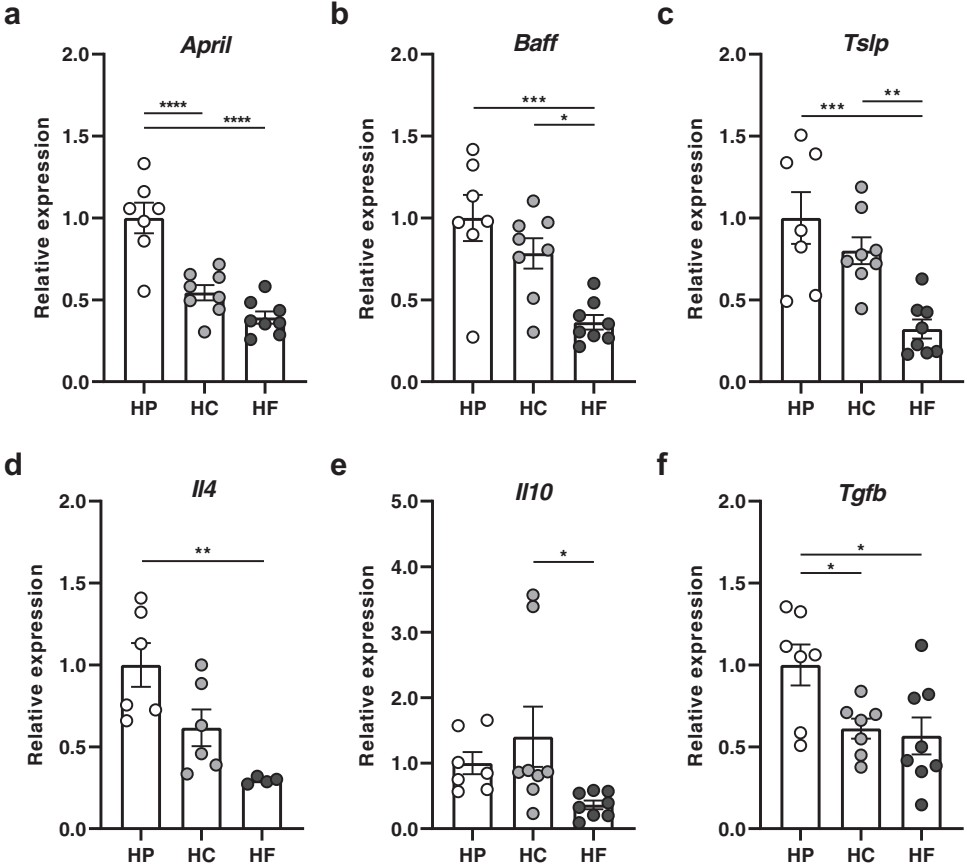

**Fig. 3 High protein feeding promotes a pro-IgA cytokine environment, specifically APRIL in the lamina propria.** Mice were fed on either a high-protein (HP), high-carbohydrate (HC), or high-fat (HF) diet for 6 weeks and intestinal ileal gene expression of **a** *April* (HP vs. HC $p = <0.0001$, HP vs. HF $p = <0.0001$), **b** *Baff* (HP vs. HF $p = 0.0005$, HC vs. HF $p = <0.0135$), **c** *Tslp* (HP vs. HF $p = 0.0005$, HC vs. HF $p = 0.0082$), **d** *Il4* (HP vs. HF $p = 0.0033$), **e** *Il10* (HC vs. HF $p = 0.0481$) and **f** *Tgfb* (HP vs. HC $p = 0.0481$, HP vs. HF $p = 0.0214$) was determined by qPCR (**a–c**, **e** $n = 7$ and $n = 8$ mice per diet for HP and HC/HF diet respectively, **d** $n = 6$ and $n = 4$ mice per diet for HP/HC and HF diet respectively, **f**; $n = 7$ and $n = 8$ mice per diet for HP/HC and HF diet respectively). Data are represented as mean ± SEM. Results represent $n = 2$ independent experiments. *$p < 0.05$, **$<0.01$, ***$<0.001$, ****$<0.0001$ by ordinary one-way ANOVA followed by Tukey's multiple comparisons test.

Fig. 4a), confirming that an HP diet promotes sIgA production via TLR4-dependent mechanisms.

Increased TLR4 signalling may be attributable to either higher expression of TLR4, increased concentration of its ligand, or increased exposure to its ligand through defects in the mucus layer or tight junction proteins. Mice fed on an HP, HC or HF diet had similar expression of TLR4 in the small intestine (Supplementary Fig. 4b), as well as comparable bacterial loads as determined by DNA concentration and total 16 S copy number (Supplementary Fig. 4c, d). We also quantified the levels of endotoxin in each diet and found low levels of endotoxin, which were comparable between the different diets (EU/ml of 0.0137, 0.053 and below the detection limit for HP, HC and HF diet, respectively). Furthermore, the expression of gut barrier-related markers *Tjp1* (tight junction protein 1) and *Ocln* (occludin) were similar between groups (Supplementary Fig. 4e), as was the integrity of the mucus layer (Supplementary Fig. 4f) and *Muc2* (mucin 2) expression (Supplementary Fig. 4g). These results indicate that increased activation of TLR4 and sIgA production under HP-feeding conditions are not linked to impaired gut integrity, higher bacterial load or higher *Tlr4* expression.

Typically, the mucus layer mediates an efficient physical separation of microbes from the gut epithelium. We, therefore, hypothesised that in vivo activation of TLR4 might be mediated by smaller structures containing PAMPs, rather than through physical interaction with whole bacteria. Bacterial extracellular vesicles (EV)

are small spherical structures less than 300 nm in diameter, produced by the budding of the membrane and consequently contain PAMPs. EV can contain nucleic acid, protein, metabolites and other molecules as cargo and are a key component of bacterial communication[21]. EV derived from *Bacteroides fragilis* has been shown to promote regulatory T-cell differentiation through the stimulation of TLR[22], highlighting the ability of EV to traverse the mucus layer to reach the host. As such, we investigated whether EV derived from the microbiota of HP-fed mice might activate TLR and promote cytokines involved in IgA CSR.

To determine whether the small intestinal environment of HP-fed mice could differentially activate TLR4 signalling, we used the reporter cell line, HEK-Blue mTLR4, in which the intensity of TLR4 activation is assessed through a colorimetric assay. These cells were incubated with the non-bacterial fraction of small intestine luminal contents isolated from mice fed on HP, HC or HF diets. HEK cells stimulated with HP small intestine content had significantly higher levels of TLR4 activation than those stimulated with either HC or HF small intestine content (Fig. 5a), suggesting the highest potential for HP-fed mice to activate host cells.

The luminal environment contains bacterial metabolites, food derivatives and bacterial EV, but bacterial EV are the only candidates containing bacterial motifs able to activate TLR4. Thus, we characterised the number and size distribution of microbiota-derived EV from mice fed on different diets by nanoparticle tracking analysis (NTA). While microbiota-derived

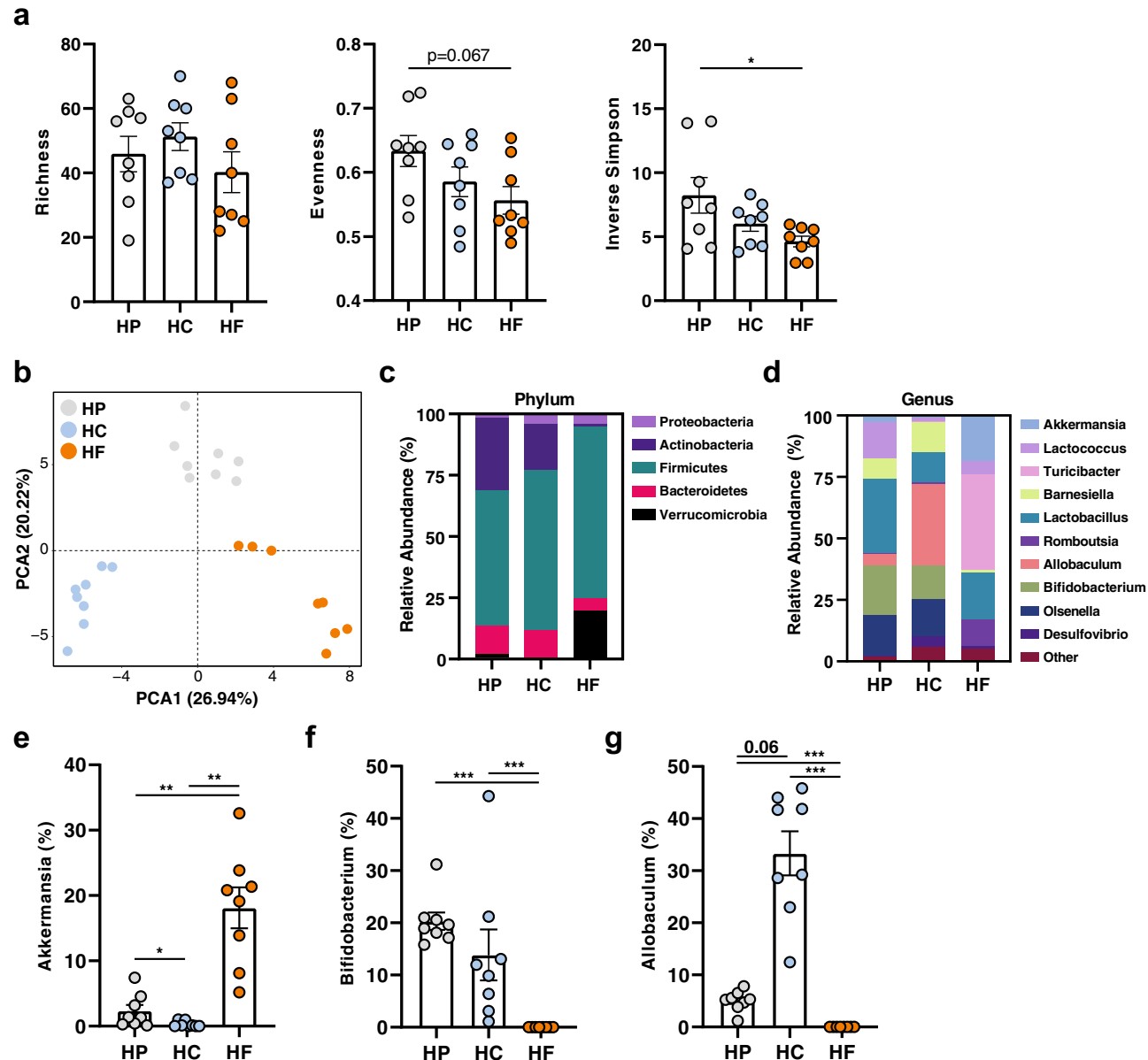

**Fig. 4 Dietary intervention significantly affects the small intestine microbiota composition.** Mice were fed on either a high-protein (HP), high-carbohydrate (HC), or high-fat (HF) diet for 6 weeks and DNA was extracted from small intestinal content for 16 S rRNA gene sequencing ($n = 8$ mice per diet). Diversity of the small intestinal microbiome was determined by **a** Richness, Evenness and Inverse Simpson Index (HP vs. HF $p = 0.0269$) with *$p < 0.05$ by ordinary one-way ANOVA followed by Tukey's multiple comparisons test. **b** Differences in the structure of the small intestinal microbiota communities were determined by principal component analysis (PCA) of Aitchison's distance (Euclidean distance of centre-log transformed counts). Relative abundance of bacteria in the small intestine is represented at the **c** phylum and **d** genus level (showing only the top 10 genera). Relative abundance of (**e**) *Akkermansia* (HP vs. HC $p = 0.032/p = 0.013$, HP vs. HF $p = 0.002/p = 0.003$, HC vs. HF $p = 0.002/p = 0.003$), **f** *Bifidobacterium* (HP vs. HF $p = <0.0001/p = 0.001$, HC vs. HF $p = <0.0001/p = 0.001$), and (**g**) *Allobaculum* (HP vs. HF $p = <0.0001/p = 0.001$, HC vs. HF $p = <0.0001/p = 0.001$) with, **<0.01, ***<0.001 by Aldex2 test (Welch's *t* test/Wilcoxon test with Benjamini–Hochberg corrected false discovery rate). Data are represented as mean ± SEM.

EV isolated from the small intestine luminal content had a similar size distribution across the diets (Fig. 5b), we found a 2-fold increase in concentration under HP feeding conditions (Fig. 5b). To determine whether microbiota-EV isolated from mice fed on the different diets differentially activated TLR4, we incubated HEK-Blue mTLR4 cells with similar doses of small intestine microbiota-derived EV relative to those observed in vivo (2:1:1 ratio of HP:HC:HF). Consistent with previous results showing the highest potential of small intestine content to activate TLR4, we found that purified EV derived from HP microbiota stimulated TLR4 and to the highest extent (Fig. 5c). However, when the

HEK-Blue mTLR4 cells were incubated with the same concentration of EV from each group, the activation of TLR4 was similar (Fig. 5c). Thus, the higher activation of TLR4 by microbiota-derived EV under HP feeding conditions was due to a quantitative rather than a qualitative effect.

To determine the impact of microbiota-derived EV on host sIgA production and translocation, we quantified the gene expression of *CCL28, APRIL* and *PIGR* in HT-29 cells incubated with vehicle control (PBS) or with small intestine microbiota-derived EV from mice fed on HP, HF and HC at physiological levels (HP > HC = HF). EV derived from HP small intestine

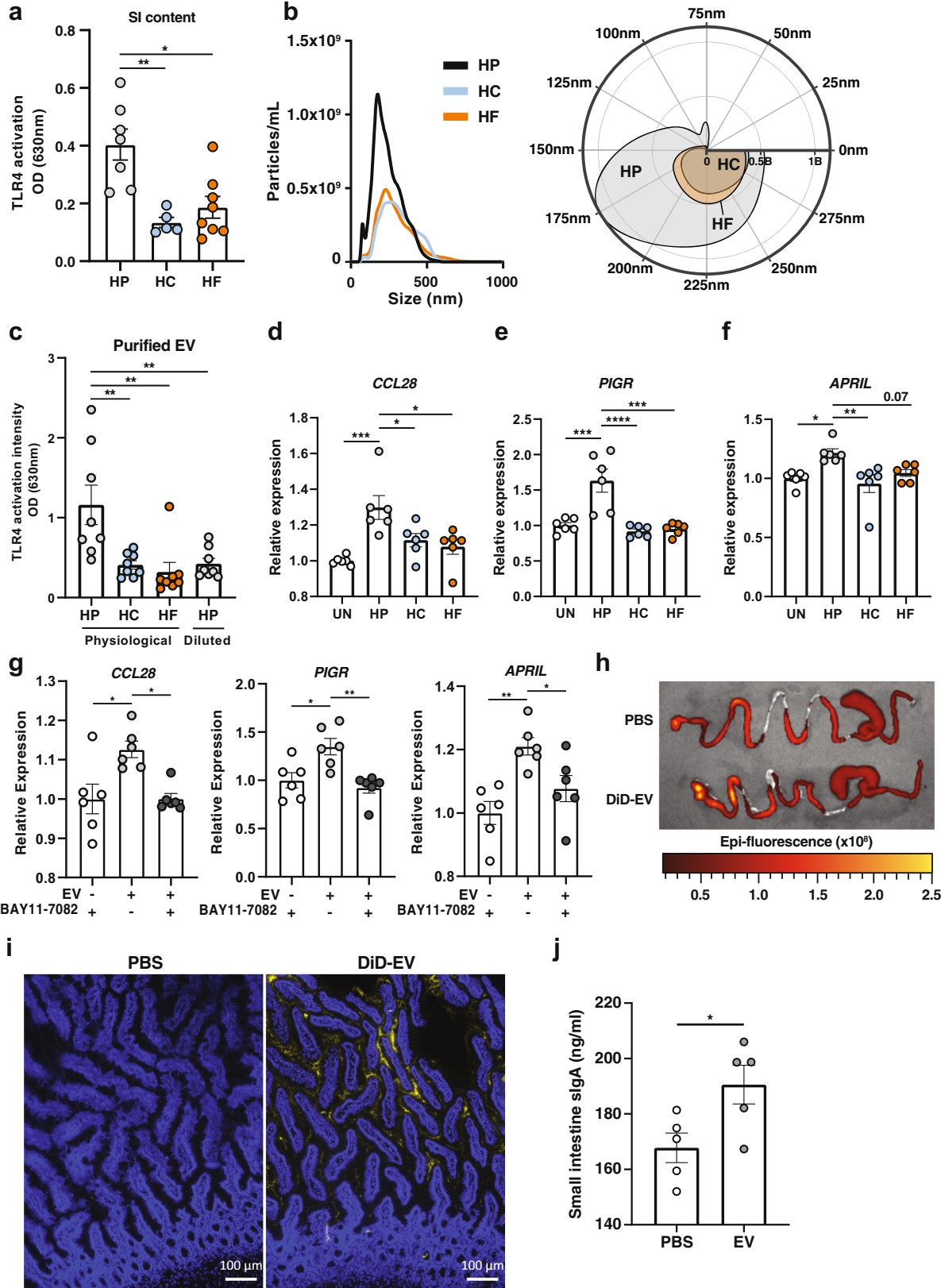

microbiota significantly upregulated *CCL28, PIGR and APRIL*, compared to cells treated with PBS or EV derived from HC and HF microbiota, and close to significance for APRIL when compared to EV derived from HC microbiota (Fig. 5d–f). To confirm whether HP-derived EV mediated these effects via TLR

signalling, we cultured HT-29 cells with EV in the presence of BAY11-7082, an inhibitor of NF-κB, the transcription factor involved in TLR signalling[12]. The addition of BAY11-7082 abrogated the effect of HP-EV on the expression of *PIGR, APRIL* and *CCL28* (Fig. 5g), suggesting that EV mediated their effects via

**Fig. 5 EV derived from high protein-fed microbiota activate epithelial TLR4 and promote the expression of PIGR and APRIL.** Mice were fed on either a high-protein (HP), high-carbohydrate (HC) or high-fat (HF) diet for 6 weeks. **a** Small intestinal content (diluted at 1:200) ($n = 5$ and $n = 8$ mice per diet for HP/HC and HF diet, respectively) was incubated overnight with HEK-TLR4 cell line and TLR4 activation was quantified at 630 nm. (HP vs. HC $p = 0.0017$, HP vs. HF $p = 0.0038$). **b** Small intestine microbiota-derived EV were characterised by Nanoparticle Tracking Analysis ($n = 2$ per diet pooled from $n = 4$ mice each) and represented as an XY plot (left): 0–500 nm vs. particle number/mL, or polar plot (right): angular axis represents particle size between 0 and 300 nm and radial axis represents particle number/mL). **c** Small intestine microbiota-derived extracellular vesicles (EV) were incubated at physiological ratio (~2:1:1 of HP:HC:HF), or at 1:1:1 ratio (HP diluted) overnight with the HEK-Blue TLR4 cell line and TLR activation measured at 630 nm ($n = 8$ mice per condition; HP vs. HC $p = 0.0058$, HP vs. HF $p = 0.0019$, HP vs. HP diluted $p = 0.007$). **d**–**f** HT-29 were stimulated with small intestine microbiota-derived EV isolated from HP ($2 \times 10^9$ EV per well) or HC/HF ($1 \times 10^9$ EV per well) fed animals for 16 h and expression of **d** CCL28 (Un vs. HP $p = 0.0006$, HP vs. HC $p = 0.0372$, HP vs. HF $p = 0.0105$), **e** PIGR (Un vs. HP $p = 0.0003$, HP vs. HC $p = <0.0001$, HP vs. HF $p = 0.0001$) and **f** APRIL (Un vs. HP $p = 0.0165$, HP vs. HF $p = 0.0036$) were quantified by qPCR ($n = 6$ wells per condition). **g** HT-29 cells were incubated with $2 \times 10^9$ small intestine microbiota-derived EV from HP-fed animals for 16 hours in the presence or absence of 10 μM of NF-κB inhibitor BAY11-7082, or BAY11-7082 alone and expression of CCL28 (−EV + BAY11-7082 vs. +EV-BAY11-7082 $p = 0.0103$, +EV-BAY11-7082 vs. +EV + BAY11-7082 $p = 0.0106$), PIGR (−EV + BAY11-7082 vs. +EV-BAY11-7082 $p = 0.0131$, +EV-BAY11-7082 vs. +EV + BAY11-7082 $p = 0.0032$) and APRIL (−EV + BAY11-7082 vs. +EV-BAY11-7082 $p = 0.0021$, +EV-BAY11-7082 vs. +EV + BAY11-7082 $p = 0.0443$) were quantified by qPCR ($n = 6$ wells per condition). **h** Representative ex vivo epifluorescence imaging of gastrointestinal tract 6 hours post-oral administration of DiD-stained EV or PBS. **i** Presence of DiD-EV (yellow) and DAPI (blue) were assessed by fluorescent microscopy from sections of small intestine isolated from mice 6 hours after oral administration, with PBS as control. Scale bar represents 100 μm. **j** Mice were intragastrically administered with $1–3 \times 10^{10}$ EV daily for 5 weeks and small intestine sIgA quantified by ELISA ($n = 5$ mice per group; $p = 0.0319$ by two-tailed unpaired t test). Data are represented as mean ± SEM. Results represent $n = 2$ independent (**a**–**h**) and $n = 1$ independent experiment (**i**, **j**). *$p < 0.05$, **<0.01, ***<0.001, ****<0.0001 by ordinary one-way ANOVA followed by Tukey's multiple comparisons test unless otherwise stated.

TLR signalling. We confirmed these in vitro findings using caecum microbiota-derived EV (Supplementary Fig. 4h–k), a site containing a higher density and purity of bacteria.

To determine whether microbiota-derived EV could reach the host small intestine epithelium in vivo, we isolated microbiota-EV from mice fed on normal chow, fluorescently labelled them with DiD, and administered them by gavage to another set of mice. Ex vivo imaging of gut tissue by IVIS revealed higher epifluorescence intensity in the gastrointestinal tract of mice receiving DiD-EV, compared to the PBS control, showing that DiD-EV readily reached the small intestine (Fig. 5h). There was also a presence of DiD-EV in both the lumen and mucosa of these mice, as determined by fluorescence microscopy (Fig. 5i). Finally, we showed that daily administration of purified microbiota-derived EV for 5 weeks in vivo could increase sIgA levels (Fig. 5j).

Together, these data highlight a close interaction between bacterial EV and host cells in vivo as well as a role for gut microbiota-derived EV on the regulation of host genes involved in sIgA regulation via TLR activation.

**Succinate promotes bacterial ROS and bacterial EV production.** In addition to TLR activation, the gut microbiota can also modulate host sIgA production through the generation of metabolites, such as short-chain fatty acids[23–25]. To determine whether such mechanisms are involved, we quantified microbiota-derived metabolites by NMR and focused on metabolites that are significantly increased under HP feeding conditions (Supplementary Tables 10-11). Of the major bacterial metabolites detected, we found that succinate was significantly elevated in both the small intestine luminal content (Fig. 6a), and caecal content (Supplementary Fig. 5a) in HP-fed mice, compared to mice fed on HC and HF diets. Of note, succinate was not detectable in the diets used in this study (Supplementary Fig. 5b). We incubated HT-29 cells with succinate to determine whether it could directly affect CCL28, APRIL and PIGR expression. We found that, unlike EV-stimulated HT-29, succinate could not directly induce the expression of CCL28, APRIL or PIGR expression (Supplementary Fig. 5c). As such, we hypothesised that changes to the gut metabolite environment would affect the gut microbiota, rather than the host directly to elicit host sIgA response.

Like any cell types, bacteria can produce EV in response to stress signals such as reactive oxygen species (ROS)[21]. Succinate,

which we found to be highly upregulated in HP-fed mice, has been shown to increase ROS in host cells such as macrophages[26]. High levels of microbiota-produced succinate under HP-feeding conditions might promote bacterial ROS, which would, in turn, stimulate the release of EV. To test this hypothesis, we incubated E. coli with increasing concentrations of succinate and quantified ROS. Strikingly, succinate increased ROS production in a dose-dependent manner (Fig. 6b) without affecting bacterial growth (Supplementary Fig. 5d). To determine the impact of succinate on EV production, we cultured E. coli with succinate for 16 h and found that succinate significantly increased EV production by NTA at the high concentrations (Fig. 6c). These results uncover a mechanism through which gut bacterial vesiculation is regulated by the metabolites in the environment, particularly succinate.

**High bacterial EV and high succinate production under HP feeding correlates with worse DSS-induced colitis.** Increased succinate has also been reported in human inflammatory bowel disease (IBD), as well as in DSS-induced colitis in mice[27,28]. Accordingly, we found that mice treated with DSS had higher gut bacterial EV production with similar size distribution (Fig. 6d, e). As observed under HP feeding conditions, IBD is not only characterised by high gut succinate levels[27] but also increased TLR activation by gut microbial PAMPs[29]. We found that DSS colitis was exacerbated under conditions of high succinate, either through HP feeding (Fig. 6f–h) or direct administration of succinate in drinking water (Fig. 6i–k). Our results suggest succinate is a key mediator that can contribute to an inflammatory intestinal environment by promoting the production of microbiota EV that can activate TLR4. This model is summarised in Fig. 7.

Together, these results highlight a potential pathway involved in IBD severity through the increased production of the bacterial metabolite, succinate, in the induction of bacterial EV that can in turn activate TLR signalling and modulate host gene expression and inflammatory responses.

**Discussion**

In the present study, we have identified a key role for dietary macronutrient composition in the dynamic regulation of IgA production and sIgA release in the small intestinal lumen. Dietary

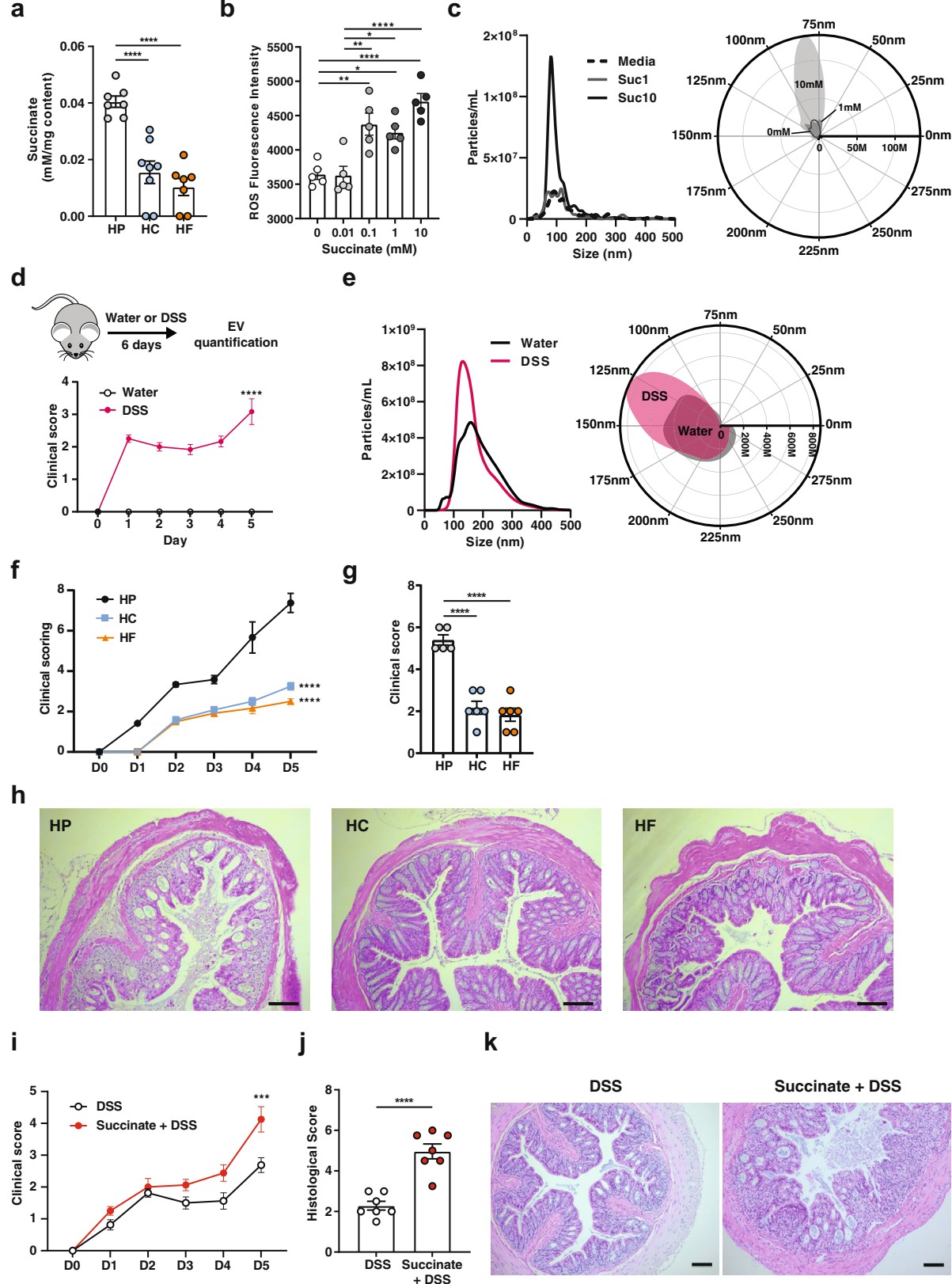

protein was the major driver of T-cell-independent sIgA response, associated with increased expression of the CSR-promoting cytokines APRIL and BAFF. Increased luminal sIgA was associated with elevated expression of the transporter *pIgR* that was correlated with the highest expression of IL-4, a cytokine involved in both IgA induction and pIgR expression. These

changes are commonly observed under pro-inflammatory conditions linked to increased activation of host pattern-recognition receptors by gut bacterial products. Accordingly, we found that intestinal luminal content from HP-fed mice activated the TLR4-NFκB pathway to a greater extent than contents obtained from either HC or HF-fed mice. This was attributed to the greater

**Fig. 6 Succinate promote bacterial ROS and bacterial EV production and is associated with worse DSS-induced colitis. a** The concentration of succinate in the small intestine luminal content of mice fed on a high-protein (HP), high-carbohydrate (HC) or high-fat (HF) diet for 6 weeks was quantified by NMR spectroscopy ($n = 7$ and $n = 8$ mice per diet for HP/HF and HC group respectively; HP vs. HC $p = $ <0.0001, HP vs. HF $p = $ <0.0001). **b** E. coli were grown in the presence of increasing concentration of succinate (0–10 mM) for 2 h and ROS production quantified by the conversion of 2',7'-dichlorofluorescein diacetate to 2',7'-dichlorofluorescein ($n = 5$ independent culture per condition; 0 vs. 0.1 mM $p = 0.0026$, 0 vs. 1 mM $p = 0.0131$, 0 vs 10 mM $p = $ <0.0001, 0.01 vs. 0.1 mM $p = 0.0021$, 0.01 vs. 1 mM $p = 0.0108$, 0.01 vs. 10 mM $p = $ <0.0001). **c** E. coli was grown in the presence of succinate (0, 1, or 10 mM) for 16 h and extracellular vesicles (EV) isolated and quantified by NTA ($n = 5$–6 per condition) and represented as XY plot (left): 0–500 nm vs. particle number/mL or polar plot (right): angular axis represents particle size between 0 and 300 nm and radial axis represents particle concentration in number/mL. **d** Mice received 3% DSS in drinking water for 6 days (upper) and colitis development was scored daily (lower) ($n = 6$ mice per group; $p = $ <0.0001 by two-way ANOVA) and (**e**) at endpoint, faecal EV were isolated and quantified by Nanoparticle Tracking Analysis ($n = 6$ mice per condition) and represented as an XY plot (left): 0–500 nm vs. particle number/mL, or polar plot (right): angular axis represents particle size between 0 and 300 nm and radial axis represents particle concentration in number/mL. **f–h** Mice were mice fed on a HP, HC, or HF diet for 6 weeks before induction of DSS colitis (3% DSS in drinking water for 6 days) and (**f**) clinical colitis development was scored daily (HP vs. HC $p = $ <0.001, HP vs. HF $p = $ <0.0001) and **g** colonic histological score assessed (HP vs. HC $p = $ <0.0001, HP vs. HF $p = $ <0.0001) and **h** representative haematoxylin and eosin-stained colonic sections ($n = 6$ mice per group). **i–k** Mice were administered 100 mM pH-adjusted succinate in drinking water for 3 weeks before induction of DSS colitis (3% DSS in drinking water for 6 days) and **i** clinical colitis development was scored daily ($p = 0.0005$ by two-way ANOVA) and **j** histological scores quantified at endpoint on colonic section ($p = $ <0.0001 by two-tailed unpaired $t$ test) with **k** representative haematoxylin and eosin-stained colonic sections shown. ($n = 7$ mice per group). Scale bar represents 100 μm. Data are represented as mean ± SEM. Results represent $n = 3$ independent (**b**), $n = 2$ (**b–e**) and $n = 1$ independent experiments (**f–k**). *$p < 0.05$, **<0.01, ***<0.001, ****<0.0001 and were analysed by ordinary one-way ANOVA followed by Tukey's multiple comparisons test unless otherwise stated.

production of microbiota-derived EV that could activate the TLR4 signalling pathway under HP feeding conditions. Increased production of EV is likely the result of increased microbial stress, as previously reported in bacteria[21]. Consistent with this, we found that high protein consumption significantly increased microbial succinate production, which we demonstrated to promote bacterial ROS production in a concentration-dependent manner in vitro. This work highlights a mechanism through which the gut microbiota regulates host sIgA response modulated by diet composition.

The majority of sIgA produced under homoeostatic conditions is induced by commensal bacteria via T-cell-independent mechanisms. Indeed, sIgA induced by the monocolonization of germ-free mice with various strains of bacteria were mostly non-specific and polyreactive in nature[4]. This observation highlights that the presence, rather than the composition of the microbiota is important in inducing sIgA and the mechanism likely involves a shared feature of bacteria, such as their capacity to activate host TLR via MAMPs. Our small intestinal microbiome data revealed no obvious dysbiosis associated with HP feeding that could explain the increase of sIgA. Rather, lower microbiota diversity was observed in HF-fed animals, an indicator of dysbiosis commonly observed in patients with allergies or IBD[30–32]. Furthermore, an HP microbiome was associated with a higher abundance of bacteria from the genus *Bifidobacterium*, commonly considered a beneficial bacteria. Additionally, phyla commonly associated with adherent intestinal bacteria, which are typically considered pro-inflammatory[33–35], such as those from *Firmicutes*, *Bacteroidetes* and *Proteobacteria* did not differ between HP- and HC-fed groups. Despite this, an HP microbiome had an exacerbated pro-inflammatory potential via increased production of microbiota EV that can activate TLR4, which was associated with exacerbated colitis compared to both HC- and HF-fed mice. Reduced colitis severity under HC feeding is likely due to increased levels of SCFA, which have been shown to be protective[36]. HF had similarly reduced colitis either due to the macronutrient composition or to the high cellulose content used as filler in this diet. While cellulose is not a functional fibre and does not yield fermentable products such as SCFA, it has been shown to be protective against colitis[37]. Future studies examining qualitative aspects of macronutrients, such as different types of fat and protein (i.e., animal vs. plant-based protein), would be required to fully appreciate the role of macronutrients on gut

homoeostasis as well as on the host sIgA responses. For example, soy-derived protein has been associated with lower sIgA, compared to casein-derived protein[38].

How the gut microbiota can adequately activate TLR under homoeostatic conditions remains unclear, as TLR is usually inaccessible to whole bacteria due to the presence of a mucus layer. We highlight that microbiota-derived EV might be the link as they can traverse the mucus and can activate TLR4 to upregulate IgA-related epithelial genes in vitro, revealing a mechanism underpinning host-microbiota mutualism. HT-29, being a human cell line, highlights the translational potential of our finding. Quantification of microbiota-derived EV could prove a useful tool to evaluate the functional characteristics as well as the inflammatory potential of the gut microbiota as an alternative to sequencing.

Our study examined both qualitative and quantitative effects of microbiota-derived EV on the host. TLR4 and CCL28 were modulated by EV in a dose-dependent manner, with HP feeding conditions increasing the activity of the former and gene expression of the latter. This suggests that EV could be a mechanism by which the host perceives changes in the state of the gut microbiota via TLR activation. As a result, the host would mobilise B cells to the gut and increase the translocation of sIgA to control the situation based on the amount of EV sensed. This could have clinical implications in diseases such as food allergy, in which low levels of IgA are commonly reported. We also uncovered a qualitative effect of EV on the host. Significant changes in the gut microbiota composition have been shown to promote the production of different types of EV. Indeed, EV can carry a range of molecules including proteins, metabolites and nucleic acids as cargo, and EV have been shown to mediate the virulence and pro-inflammatory effects of pathogenic bacteria[21]. A complete analysis of the features of EV derived from HP, HC and HF microbiota, although outside the scope of this study, would confirm this.

The high levels of succinate under HP-feeding conditions remain unexplained. We could not identify a specific microbiome signature that would account for greater succinate production in HP-fed mice. One possibility is the disappearance of succinate consumers in the HP microbiota. We find that succinate could modulate ROS production in bacteria and the long-term impact of this on the gut microbiota is yet to be determined. ROS is known to induce bacterial mutation and antibiotic resistance which could thus raise concerns that high protein diet feeding could promote the potential growth of antibiotic-resistant

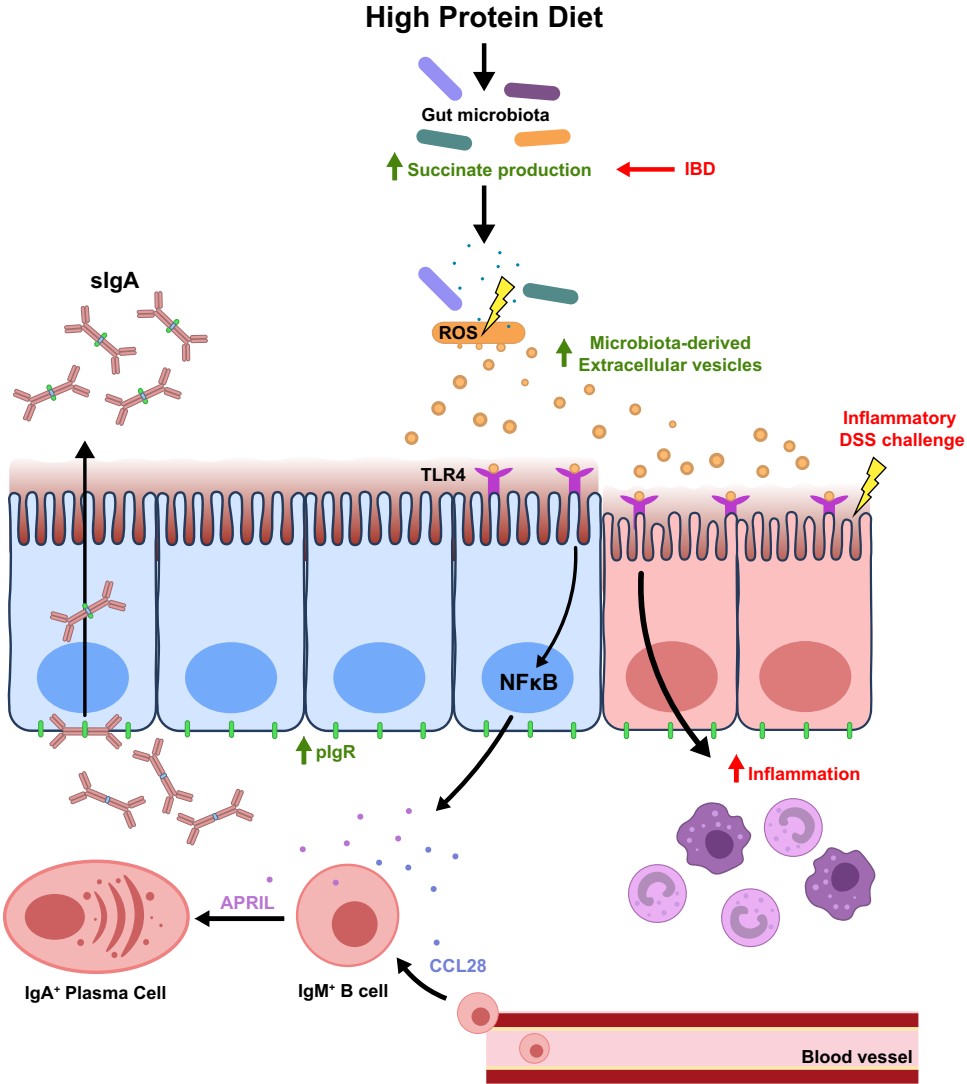

**Fig. 7 Model of high protein diet in the induction of sIgA response.** High protein diet feeding promotes succinate production by the gut microbiota. High levels of luminal succinate induce gut bacterial cellular stress and reactive oxygen species (ROS) production, which promote vesiculation and increased production of microbiota-derived extracellular vesicles. Microbiota-derived extracellular vesicles can directly activate TLR4 expressed on the gut epithelium, activating downstream NFκB signalling that results in increased expression of APRIL, CCL28 and PIGR, which drives the T-cell-independent sIgA response. Increased TLR4 signalling also potentiates pro-inflammatory responses, resulting in more severe DSS-induced colitis. Increased intestinal succinate observed in IBD patients may contribute to disease pathology by promoting microbiota-derived extracellular vesicle production and the TLR4/NFκB signalling pathway.

bacteria. Moreover, we identified higher levels of bacterial EV in DSS colitis, a disease in which gut bacterial dysbiosis, as well as high succinate levels, were previously reported[27,28]. This result suggests that dysbiosis, in general, could be sensed by the host via EV. However, whether bacterial dysbiosis observed in diseases other than IBD is associated with higher microbiota EV production remains unknown.

In summary, we describe a pathway by which the gut microbiota can regulate T-cell-independent IgA induction via bacteria-derived EV as illustrated in Fig. 7. More broadly, we show a mechanism through which the microbiota impact the host. This opens exciting new prospects for characterising microbiota EV as a biomarker for dysbiosis, as well as the potential use of microbiota EV as a postbiotic to restore gut homoeostasis.

## Methods

**Animals and housing**. Male C57BL/6 mice (6 weeks of age) were purchased from Animal Bioresource (NSW, Australia). Mice were maintained under specific pathogen-free conditions at 22 °C, 50% humidity with a 12 hour light/dark cycle (6PM–6AM) in the animal facility of the Charles Perkins Centre. Animals were housed in cages with non-edible compressed cotton fibre (iso-PADS) bedding. All experiments were performed in accordance with protocols approved by the University of Sydney Animal Ethics Committee (Protocol 2017/1280, 2019/1493, and 2019/1688).

**Animal diets and feeding**. Animals were fed *ad libitum* on either a high protein, moderately low carbohydrate, moderately low-fat diet (P60, C20, F20; henceforth, HP); a low protein, high carbohydrate, moderately low-fat diet (P5, C75, F20; henceforth, HC), or a low protein, moderately low carbohydrate, high-fat diet (P5, C20, F75; henceforth, HF). These three diets represented the extremes (apices) from a larger set of 10 diet compositions (Supplementary Table 1) that were used in a subset of the experiments, encompassing a macronutrient range of protein (5–60%), carbohydrate (20–75%), and fat (20–75%) chosen using nutritional geometry to comprehensively sample dietary macronutrient mixture space[39]. All diets were isocaloric at 14.5 MJ/kg and based on a modification to the semi-purified AIN93G formulation and purchased from Specialty Feeds, Glenn Forest, Australia (SF17-188-SF17-197) and the HP/HC/HF used for most experiments had very low endotoxin levels as tested with the PyroGene Recombinant Factor C kit (Lonza).

**Quantification of immunoglobulins**. Small intestinal content was resuspended in PBS (100 mg/ml) containing cOmplete™ Protease Inhibitor Cocktail (Roche), centrifuged at $14,000 \times g$ for 10 min to pellet bacteria and debris and supernatant collected. Quantification of IgA and IgM levels was performed using the Mouse IgA or IgM ELISA quantification set (Bethyl Laboratories), or the Mouse IgA ELISA Antibody Pair Kit (STEMCELL Technologies) according to the manufacturer's instruction.

**Immunofluorescence**. For immunohistochemistry, frozen small intestine sections were fixed with ice-cold methanol, and incubated with goat anti-mouse IgA (1:100, Bethyl Laboratories) followed by AF488-conjugated chicken anti-goat IgG secondary antibody (1:200, Invitrogen). Sections were stained with DAPI and examined with an Olympus BX51 Microscope and images were captured using the cellSens Standard v1.12 software.

**Flow cytometry**. Single-cell suspensions were prepared by mechanical disruption of tissues through a 100 µm filter. For isolation of intestinal lamina propria cells, mesenteric fat and Peyer's patches were first resected from small intestine tissues, then tissues washed with PBS and cut into 1 cm pieces and incubated in pre-digestion buffer (HBSS containing 10% FBS, 10 mM HEPES, 5 mM EDTA) for 40 min at 37 °C at 200 rpm in an orbital incubator. Tissues were washed twice with PBS then cut into <1 mm pieces and incubated in digestion buffer (RPMI containing 10% FBS, 10 mM HEPES, 2.5 mg/ml collagenase type IV (Gibco)) for 60 min at 37 °C at 200 rpm in an orbital incubator and lymphocytes were enriched via a 40/80% Percoll gradient (GE Healthcare). Cells were incubated with LIVE/DEAD™ Fixable Blue Dead Cell Stain Kit (ThermoFisher Scientific) for exclusion of non-viable cells and anti-mouse CD16/32 (91; BioLegend) for blocking non-specific binding, then stained with the antibodies described in Supplementary Table 12 and acquired on the LSRII (BD Bioscience) with the BD FACS Diva v8.0.2 software. Data were analysed with the FlowJo v10.4.2 software (FlowJo LLC).

**RNA extraction and qPCR**. Total RNA was extracted using TRI Reagent (Sigma-Aldrich) following the manufacturer's instructions and cDNA was generated using the High-Capacity cDNA Reverse Transcription Kit (ThermoFisher Scientific). qPCR was conducted using the Power SYBR™ Green PCR Master Mix (Thermo-Fisher Scientific) on a LightCycler® 480 Instrument II (Roche Life Science) and LightCycler® 480 v1.5.0 software: 1 cycle of 95 °C for 5 min; 45 cycles of 95 °C for 15 s, 60 °C for 60 s followed by dissociation curve: 95 °C for 5 s, 65 °C for 60 s followed by 0.11 °C/s to 97 °C. Expression was normalised to a housekeeping gene as indicated in the Figure legends. Primers were used at a final concentration of 200 nM, and sequences are provided in Supplementary Table 13.

**16 S rRNA gene sequencing and bioinformatics**. DNA from small intestinal jejunum content was extracted using the FastDNA™ SPIN Kit for Feces (MP Biomedicals). Illumina sequencing of the V4 region (515f-806r) of the 16 S rRNA gene was performed commercially at the Ramaciotti Centre for Genomics (UNSW, Australia). Paired-end reads ($2 \times 250$ bp) were processed with the dada2 package (1.12.1) using R software (3.6.1) with RStudio (1.4.1717) to generate amplicon sequence variant (ASV)[40]. Taxonomic classification of ASV was performed using the Ribosomal Database Project naïve Bayesian classifier (rdp_train_set_16) with species-level taxonomy assignment (doi:10.5281/zenodo.801828). For downstream analysis, ASV with a total abundance of less than 0.01% was first filtered out. For beta diversity analysis, a compositional data analysis approach was used[41]. ASV was centre log-ratio (CLR) transformed after replacement of zeros via count zero multiplicative replacement[42]. Differences in microbiome composition between 2 groups were determined by PERMANOVA of Aitchison distance (Euclidean distance of CLR transformed data), after validation of homogeneity of group dispersions. For differential abundance testing of ASV between two groups, ALDEx2 was used, and taxa were considered differentially abundant when Benjamini–Hochberg corrected the expected $p$ value for both Welch's $t$ test and Wilcoxon test were <0.05 and magnitude of effect size >1. Analysis was performed with the following R packages phyloseq (1.32.0), vegan (2.5.7), ALDEx2 (1.20.0), zCompositions (1.3.4), ranacapa (0.1.0). DNA sequencing data were deposited in the European Nucleotide Archive under accession number PRJEB39583.

**Quantitative assessment of intestinal bacterial load**. Intestinal bacterial load was determined by absolute 16 S copy number per mg of intestinal content, determined by qPCR using the following 16 S rDNA universal primers: UniF: GTGSTGCAYGGYYGTCGTCA and UniR: ACGTCRTCCMCNCCTTCCTC with the following cycling protocol: 1 cycle of 95 °C for 5 min; 45 cycles of 95 °C for 30 s, 60 °C for 30 s, 72 °C for 30 s, 1 cycle of 72 °C for 5 min followed by dissociation curve: 95 °C for 5 s, 65 °C for 60 s followed by 0.11 °C/s to 97 °C. Copy number was determined via a standard curve ($10^8$ to $10^2$ copies/µl).

**EV isolation and nanoparticle tracking analysis (NTA)**. In all, 100 mg of caecum content or small intestine luminal content was homogenised in 0.02 µm filtered PBS and EV isolated by differential centrifugation, as described previously[43,44]. Briefly, the homogenate was centrifuged at $340 \times g$ for 10 min at 4 °C. Supernatant

was then centrifuged at $10,000 \times g$ for 20 min at 4 °C followed by $18,000 \times g$ for 45 min at 4 °C, filtered through a 0.22 µm filter and centrifuged at $100,000 \times g$ for 2 hr at 4 °C. EV pellet was resuspended in 0.02 µm filtered PBS. EV particle size and concentration were assessed by NTA on a Nanosight NS300 (Malvern Instruments Limited). NTA 3.2 software was used to calculate EV concentration and size distribution.

**Ex vivo imaging of EV uptake**. Fluorescent staining of EV was performed with the Vybrant DiD Cell Labelling Kit (Invitrogen) at a final concentration of 5 µM for 30 min at room temperature and washed with PBS by ultracentrifugation at $100,000 \times g$ for 2 h at 4 °C. Animals were intragastrically administered 80 µg of DiD-EV and organs imaged 6 h later with the IVIS Spectrum In Vivo Imaging System (Perkin Elmer) at excitation: 640 nm and emission: 680 nm. Images were analysed using the Living Image 4.5 Software (Perkin Elmer) and fluorescence was quantified as radiant efficiency.

**Mucus layer thickness quantification**. For quantification of mucus thickness, colon tissues were fixed in Carnoy's solution fixative for 24 hours and then transferred to 70% ethanol prior to paraffin embedding. All samples were transversely sectioned at 7 µm and stained with Alcian blue (pH 2.5) and 0.1% (w/v) acetic safranin solutions for histological characterisation of the mucus layer. All slides were imaged by the Aperio AT2 Slide scanner at ×20 magnification and visualised using Aperio Image-Scope v12.3.3 (Leica Biosystems Imaging). Analysis was performed using the image processing software FIJI (v1.52p). The scale bar of each sample was used to determine the known length using the line tool. A total of 10 measurements of the mucus thickness were recorded per sample and values were averaged.

**TLR4 activity assay and cell culture**. HEK-Blue mTLR4 reporter cell line (InvivoGen) was maintained in complete DMEM (Gibco) containing 10% FBS (Bovogen Biologicals), 100 U/ml Penicillin and 100 µg/ml Streptomycin (Gibco), 2 mM L-glutamine (Sigma-Aldrich) and supplemented with 100 µg/ml Normocin and 1× HEK-Blue selection (InvivoGen). TLR4 activation by bacterial-derived EV or diluted small intestinal content was assessed according to the manufacturer's instructions. HT-29 cells (ATCC) were maintained in complete DMEM media and stimulated with small intestinal content or bacteria-derived EV for 16 h. NF-κB inhibitor BAY11-7082 (Sigma-Aldrich) was used at 10 µM.

**Detection of polar metabolites**. For detection of polar metabolites by $^1$H nuclear magnetic resonance spectroscopy (NMR), caecal or small intestinal content was homogenised in deuterium oxide (Sigma-Aldrich) at a concentration of 100 mg/ml and centrifuged at $14,000 \times g$ for 5 min at 4 °C. For dietary metabolite analysis, diet pellets were suspended in 5 ml deuterium oxide per gram diet and allowed to fully dissolve overnight at 4 °C before clarification by centrifugation at $14,000 \times g$ for 5 min at 4 °C. Contents were then filtered through a 3 kDa centrifugal filtration unit (Merck Millipore), and polar metabolites were collected from the aqueous phase of a chloroform-D/methanol-D/water extraction. The resulting solution was diluted with sodium triphosphate buffer (pH 7.0) containing 0.5 mM 4,4-dimethyl-4-silapentane-1-sulfonic acid as internal standard (All from Sigma-Aldrich) and samples were run on a Bruker 600 MHz AVANCE III spectrometer. The Chenomx NMR Suite v8.4 software (Chenomx Inc) was used to identify and fit compounds from the acquired sample spectrum against pre-defined spectral reference libraries and concentrations quantified in reference to the internal standard.

**Bacterial culture and ROS quantification**. Escherichia coli (K-12 MG1655) were cultured in Luria-Bertani broth with succinate, propionate or pyruvate at indicated doses and supernatants were collected for EV quantification 16 h later. For quantification of ROS production, E. coli were treated with succinate at indicated doses for 2 h, and cells pelleted by centrifugation (5000 rpm for 5 min). Pellet was resuspended in PBS containing 10 µM of 2',7'-dichlorofluorescein diacetate (Sigma-Aldrich) and incubated for 60 min at room temperature. Fluorescence intensity was measured using a TECAN Infinite M1000 microplate reader (Excitation 485 nm, Emission 538 nm, Gain 100) using the Tecan i-control v1.10.4.0 software.

**Experimental model of colitis**. Murine model of colitis was induced by the administration of 3% dextran sodium sulfate (DSS, MW: 36,000-50,000, MP Biomedicals) in drinking water for 6 days. In some experiments, mice were treated with pH-adjusted sodium succinate (100 mM, Sigma-Aldrich) in drinking water for 3 weeks prior to DSS model induction and throughout the model. Mice were monitored daily and clinical score was assessed as follows: 0-normal faeces; 1-soft faeces; 2-pasty faeces; 3-liquid faeces; 4-moderate rectal bleeding; 5-severe rectal bleeding; 6-haemorrhagia/diarrhoea and 7- signs of morbidity as previously reported[36]. Histological analysis of haematoxylin and eosin-stained colon sections were performed blinded by at least 2 independent researchers for tissue damage: 0, no mucosal damage; 1, lymphoepithelial lesions; 2, surface mucosal erosion and 3, extensive mucosal damage, extension into deeper structure, and for inflammatory cell infiltration: 0, occasional cell infiltrate; 1, increased number of infiltrating cells; 2, confluency of inflammatory cells extending to the submucosa and 3, transmural extension of the inflammatory cells.

**Mixture model and statistical analyses**. The effects of dietary macronutrient composition on outcomes were analysed using mixture models (also known as Scheffé's polynomials). Analyses were performed on each outcome variable separately. Models were implemented using the *mixexp* package (1.2.5) using R software (3.6.1). For each outcome, four models equivalent to equations described by Lawson and Willden[17] as well as a null model were fitted. Collectively, these models test for no effect, linear effects, and non-linear effects of macronutrient composition on the outcome variable. The most appropriate model was determined by Akaike information criterion (AIC), where the lowest AIC value (where AIC was within two points of difference, the simpler model was selected) was deemed the most appropriate model. Predictions from the best model were represented on a right-angled mixture triangle plot[15]. For comparison between groups, analysis was done with GraphPad Prism v9 using Ordinary one-way ANOVA followed by Tukey's multiple comparisons test. No statistical method was used to predetermine sample size. Differences were considered as significant when $p < 0.05$ where $*p < 0.05$, $**p < 0.01$, $***p < 0.001$, and $****p < 0.0001$.

**Reporting summary**. Further information on research design is available in the Nature Research Reporting Summary linked to this article.

## Data availability

The 16 S rRNA sequencing data generated in this study have been deposited in the European Nucleotide Archive database under accession code PRJEB39583. Source data are provided in this paper.

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

## Acknowledgements

This project was funded by the Australian Research Council grant APP160100627 and APP210102943 and by the Sydney Medical School MCR BioMed-Connect Grants. We thank Alexandra Angelatos for her help with some aspects of the project. We thank professor Arthur Conigrave for his input and discussion. We acknowledge the Sydney Cytometry facility for providing access to flow cytometer analysers, the Sydney Preclinical Imaging facility for training and access to the IVIS Spectrum Imaging System, and the Laboratory Animal Services at The University of Sydney for animal housing and husbandry.

## Author contributions

J.T. performed most of the experiments, participated in the project design and wrote the manuscript, D.N., J.Taitz., G.V.P., R.E., E.S., H.W., and S.J.C. participated in the experiments, A.S. and M.R. helped with the data analysis, J.W., R.N., N.J.C.K., G.E.G.,

and S.J.S. helped with the study design, L.M. participated to the study design, supervised the study, and wrote the manuscript.

## Competing interests

The authors declare no competing interests.
