## [Peer Review File · Nature Communications]

Dietary protein increases T cell independent sIgA production through changes in gut microbiota-derived extracellular vesiclesREVIEWER COMMENTS

Reviewer #1 (Remarks to the Author):

General comments:

Here, the authors propose a novel pathway through which dietary macronutrients influence sIgA production via gut microbiota-derived extracellular vesicles. This is an interesting paper that uses a range of techniques and tackles some exciting questions on the interaction between diet/microbiota/mucosal immunology. Unfortunately however, the story isn't clear and some of the experiments conducted are not sufficient to explain the proposed (complex) pathway. The experiments conducted do not sufficiently support the claims that the effect on sIgA is driven exclusively by EVs and that EV production is driven by succinate. The major methodological concern relating to this is the isolation of cecal EVs and cecal SCFA rather than EVs or metabolites produced in the small intestine. This manuscript has the potential to be very good, however the proposed pathway has too many moving parts. It would benefit from a more focused approach on parts of this pathway to rigorously confirm their effect. Detailed comments are below, which if addressed, would make this manuscript suitable for publication.

Major comments:

1. As the focus of the manuscript is on small intestine, it is unclear why EVs and metabolites are isolated from the cecum and used for all following experiments. This is a major limitation that needs to be addressed.

2. The cell line experiments appear crude and insufficient to support the story. It is not clear why HEK cells were used rather a small intestine cell line (e.g. HT-29). It is also not clear if this effect on gene expression in the cell line is purely due to a higher bacterial load in the assay. Other simple mouse experiments would be much more suited to investigate this (see some suggestions at the bottom)

3. The final DSS experiment does little to confirm the story relating to the story of protein, EVV, succinate etc.

Other comments:

1. The introduction (and the manuscript in general) is a little sparse on references. There are a number of other seminal manuscripts in the field that explore parts of this pathway with regards to sIgA production as influence by the gut microbiota etc.

2. The results in figure 1e and a number of other figures appear to be driven by a small number of outliers. I would imagine that these are cage effects, which are commonly observed in microbiota studies. Please comment on whether this is the case, whether cage effects were adjusted for in any way, and whether potential cage effects effect the main results

3. Fig 1b is coloured by predicted sIgA concentrations as assumed by linear associations. It would be beneficial to present the R² of the model in order to indicate this. It would also be beneficial to the reader if the points representing the 10 diets were also enlarged. If possible, this panel should have scales matched to 1a in order for the graphs to be compared.

4. Could the authors comment somewhere in the manuscript as to whether total calorie/food intake differed between the 10 diet groups, particularly the HP/HC/HF groups?

5. In the sentence beginning with “For sIgA, model 1 had the most favourable AIC”, it is not clear to the reader what “model 1” is and how this would differ to model 2 or model 3 etc. If possible, rephrase to indicate e.g. the strongest model was driven by protein.
6. It is not clear in the results how long the diets were fed for. Presumably all experiments were 5 weeks as stated in methods? Please state this somewhere in first paragraph of results.
7. Why are there only 4 data points in the HF diet in Fig 2b (compared with n=8 in HC?). There appears to be a relatively strong trend in the HF diet which may have been significant if this group also had 7-8 data points
8. In Fig 2c and 2e, the percentages in the flow cytometry plots don't seem to correspond with the corresponding bar plots in 2d and 2f.
9. Again, in Fig 3c, there appears to be a cage effect probably? Please clarify if these results are driven by single cages or if they have genuine wide variation across all the animals in the group irrespective of cage
10. Some of the terms used in the microbiota results are slightly confusing. E.g. how did you determine “adequate sequencing depth”. This could be approached in methods where you can detail that samples with less than X number of reads were removed.
11. Similarly, higher diversity in a microbiome is very context specific. It is not clear if higher small intestine diversity is better, in fact it could be the opposite. It may be suitable to state that higher diversity in the stool microbiome is generally associated with health, however this should be saved for the discussion.
12. Again, the statement “suggesting a dysbiotic small intestinal microbiome” is misleading and should be removed from the results.
13. Something that should be addressed in the discussion is the luminal vs adherent small intestinal microbiota. Emerging studies are beginning to investigate this and indicated that they differ quite significantly. This could have important implications for the context of this study with regards to contact with the epithelium, sIgA production etc.
14. If stating that HF had lower evenness in the results, you must explicitly state that it was not statistically significant
15. Please report PERMANOVA results (or equivalent) to show statistics that Unifrac distances were significantly different.
16. It appears that there was a significant amount of cellulose in the high fat diet. This may have had a major effect on the gut microbiome in addition to the fat itself. Could the authors provide a line or two in the discussion about the limitations of the diets.
17. Fig 4g, I would be surprised if there were only 15 taxa at genus level. Where are the others that make up the 100% relative abundance? The top 3 or 4 can't really be seen

anyway so it may be best to present the top 10-12 and have an “others” section for all the other very low abundance taxa.

18. This following statement: “While increased Proteobacteria have been linked to inflammation” is a bit of a generalization as proteobacteria is quite a broad phylum. Gammaproteobacteria are usually more inflammatory whilst other proteobacteria classes are less so. Again, I would save such statements until the discussion as they are context specific and can't be generalized to the results.

19. Again, the statement “considered as probiotics” needs to be removed. Probiotics are defined species (see WHO definition). Avoid reference to probiotics and instead just refer to beneficial properties of these genera in the context of your study

20. “By Spearman’s correlation, we confirmed that dietary protein content was more associated to sIgA than carbohydrate and fat (Fig. 4I).” This correlation is not shown in this heatmap.

21. I am unsure if there is enough variation in the F, C and P contents of the three diets to conduct spearman correlations. Spearman are usually conducted on two continuous variables, however the F, C and P contents are not continuous outcomes. They each only have 2 values, either high (60/75%) or low (5/20%). This would be ok if you were looking across a broader scale of the 10 diets. The spearman correlations with sIgA/pIgR and microbial taxa are more suitable.

22. What was cfu/g load in small intestine? Or at least 16S copy number per ug? The effect caused by HP microbiome may be based on higher bacterial load in the small intestine. This is important

23. This approach seems a bit crude and a previous method should be cited. There are not enough details on this method e.g. concentration of small intestine inoculate that was applied to cells and the rationale?

24. Why HEK cells? Why not a small intestine cell line? Small intestine epithelium cell lines also express TLR4. HEK is also a human cell line. A mouse cell line would be more suited to a mouse microbiome. Much of this proposed pathway is also mucosally mediated. Cell lines can't replicate that. Please comment

25. Was cell count of inoculate or inoculate mass normalized beforehand in HEK experiment? Was it filtered in any way? More details/rationale needed on this experiment.

26. TLR4 was not discussed before in the intro. This needs more rational before introducing here. What was the expression of TLR4 in small intestines of mice fed the 3 diets?

27. Please provide a reference for the EV isolation protocol. I am not familiar with EV isolation, however it appears to me that other extracellular products could be present after these centrifuging steps other than just EVs (e.g. ‘free’ metabolites).

28. Why cecum-derived EV? This is a critical concern. All of the work up to this point is focussed on the small intestine. Why not use SI-derived EVs?

29. The effect of EVs on TLR4 was solely due to greater number of EVs. When the same

doses were applied, there was no difference between diets. It is not clear if this was also the case for results presented in 5d-f. Were these normalized for EV quantity?

30. Another limitation of this study is the lack of discussion about how/why EVs induce this response. What do they contain? It may be outside the scope, but needs to be addressed further in the discussion.

31. In Fig 5g, which diet are these EVs from? If from HP, it is not clear why this was conducted as there was not a clear effect of HP EVs on the expression of PIGR or APRIL compared with unstimulated controls or other diets

32. Why not isolate EVs from all three diets and gavage into different mice to then see effect on SI sIgA and TLR4/APRIL etc?

33. Were the three SCFA the only metabolites upregulated in HP or why were they chosen? More detail is needed on the NMR results and why the SCFA were chosen or if other metabolites were differentially abundant. Again, the major concern here is why NMR was conducted on cecal content rather than small intestine content. There are very few SCFA in the small intestine.

34. "We found that, unlike EV-stimulated HT-29, neither pyruvate, propionate nor succinate could directly induce the expression of CCL28, APRIL or PIGR expression" - this statement should be revised as EVs did not stimulate APRIL and only HP EVs stimulated CCL28.

35. When comparing succinate with pyruvate and propionate for E. coli EV production, why has 10mM been used for succinate but 10-fold less conc used for the other SCFA (1mM)?

36. Is the EVV result in Fig 6j normalized by microbiota cell count? Could it just be that DSS reduces cfu of small intestine microbiota

37. I am not sure how this last DSS experiment concludes the story? DSS alone could affect EV production and this does little to add to the EV/succinate story. This experiment either needs to be explored further to conclude the proposed pathway or removed entirely.

38. It would be beneficial to have a conceptual figure at the end that explains the proposed pathway. This may depend on editorial guidelines, however this complex pathway involving dietary protein, small intestine microbiota, EVs, sIgA, succinate, TLR4 etc etc is very hard to visualize and comprehend.

39. Further discussions is needed to on types of macronutrients. This study is limited by the chosen sources of macronutrients. Different sources could have highly varying effects (e.g. canola oil was use here as a fat source which is high in omega-6 fats but it is not representative of omega-3, MUFA, SFA etc. Similarly, carbohydrate sources have huge effect on microbiota whether simple sugars or complex carbohydrate structures).

40. Some alternative experiments that could help to clarify the story include:

- Treat mice (ideally germ-free or antibiotic treated) with small intestine contents of HP/HC/HF mice to demonstrate that the effect on sIgA/EVs is microbiota-dependent
- Treat mice with isolated EVs from HP/HC/HF mice and measure TLR4/sIgA/APRIL etc to

show more rigorous results in vivo rather than in HEK cells

- Induce low-dose DSS colitis and treat mice with succinate or EVs to confirm it exacerbates the clinical effect.

Reviewer #2 (Remarks to the Author):

The manuscript entitled “Dietary protein increases T cell independent sIgA production through changes in gut microbiota-derived extracellular vesicles” by Tan et al. reports some interesting observation connecting diet-IgA production and microbiota. As much as the overall idea is intriguing there are some major limitation that reduce the enthusiasm for the manuscript.

General comments:

Authors claim “In our approach, we fed mice on one of 10 different isocaloric diets varying in their macronutrient composition and quantified sIgA. By using mixture modelling, we identified that dietary protein dramatically increased luminal sIgA levels, which is associated with increased expression of CCL28 and APRIL in the small intestine”. The majority of the manuscript data report only 3 diets “Animals were fed ad libitum on either a high protein, moderately low carbohydrate, moderately low-fat diet (P60, C20, F20; henceforth, HP); a low protein, high carbohydrate, moderately low-fat diet (P5, C75, F20; henceforth, HC), or a low protein, moderately low carbohydrate, high-fat diet (P5, C20, F75; henceforth, HF). These three diets represented the extremes (apices) from a larger set of 10 diet compositions (Supplementary Table 1) that were used in a subset of the experiments, encompassing a macronutrient range of protein (5-60%), carbohydrate (20-75%), and fat (20-75%) chosen using nutritional geometry to comprehensively sample dietary macronutrient mixture space”. It sounds like an overstatement to mention results obtained using 10 diets considering that these appear just in Figure 1a, b and d and Supplementary Figure 1. Only data obtained from 15-24 mice were showed. Indeed the total mice should be 80 (8 animals for each diet). Only data belonging to three diet (5-8 mice for diet) were reported. The low sample size does hardly allow generalizing those findings. Furthermore, analytical methods and statistics applied are not well described and hard to follow.

The Methods section should contain all elements necessary for interpretation and replication of the results. Please add references when appropriate. Please add more details to permit the replication of the experiments.

As mentioned, also by authors, IgA play a non redundant role in preventing microbes interaction with intestinal epithelial cells and possibly interdigitating dendritic cells. In this context it is crucial to evaluate the mucus layer thickness and integrity, furthermore, the role of the intestinal permeability should be considered (even with a simple qPCR of TJ genes). Authors results encourage the idea that mice under different dietary regimes may better respond to acute inflammatory insult like DSS administration. Is this aspect been evaluated?

Additional minor comments:

Numerous typographical and grammatical errors are littered through the manuscript and

cause confusion over interpretation. For example L. 258: “metabolites that are significantly upregulated” Genes are upregulated; metabolites are not upregulated.

- L. 77-80: To determine how dietary macronutrients might affect sIgA and thus host-microbiota mutualism... Fig. 1a and Supplementary and Table 1 did not explicitly support the points being made by the authors.

- L. 170-172. It is difficult to interpret in terms of its statistics, and does not appear to support the points being made by the authors (e.g., Supplementary Table 5).

L. 173-179. How the diversity of the gut microbiota was estimated? The technical description in this paragraph appears jumbled. Descriptions of normalizations and other important parameters need to be included.

L. 180-197. Data were not supported by statistics. Please add the FDR values to support comparison among groups. The Authors do not show the data at the species level, why?

L. 225: How EV was purified and characterized?

L. 253: Bacterial metabolites?

Reviewer #3 (Remarks to the Author):

In this manuscript, Tan et al. report on the positive impact of high protein levels in diet on secretory IgA in the intestine. By screening mice fed for 5 weeks with 10 different purified diets that differed solely in their protein, carbohydrate and fat levels, but were comparable in all other micronutrients, in combination with mathematical modeling, high protein content in combination with low fat and low carbohydrate content was determined to be the optimal diet to increase sIgA levels in the intestine. The authors continue the manuscript by comparing a protein-rich (HP) diet with one enriched in carbohydrates (HC) and a high-fat diet (HF) and present experiments to show that the affected IgA production is T cell-independent and mucosa-specific. While the composition of the small intestinal microbiota is dramatically shifted upon feeding of the three diets, the authors claim that not the composition of the microbial community but rather their ability to release different amounts of extracellular vesicles is responsible for the increase in sIgA levels following HP feeding. By a combination of in vitro experiments with different cell lines, the authors show that HP-fed murine contents contained more EVs which are able to stimulate TLR4 on a coloncarcinoma cell line and induce IgA promoting proteins, such as APRIL and PIGR. The increased production of EVs under HP feeding seemed dependent on succinate-induced bacterial ROS production. While the overall topic of how dietary components can affect the intestinal host immune response either by themselves or through modification of the commensal microbiota is a research area of high importance to understand the host-microbial mutualism, and still under investigated, I am lacking important controls and additional experiments in this manuscript that are required to support the conclusions drawn by the authors. In my opinion, the manuscript is not acceptable for publication in Nature Communications in its present form and will need major revisions.

Please find my more detailed comments below:

1. The authors focus most of the paper on three diets with extreme protein, fat or carbohydrate content. While I can understand that testing all 10 diets in vivo is not feasible, I would have liked to see a control diet that is composed in regard to macronutrients as the regular mouse chow used in most animal facilities to have a baseline value for the read-outs. This would increase the physiological impact of the data as one could see which of the HP, HC and HP diet actually lies outside or around the “average” values in comparison to control diet.

2. Figure 1f: The IgA staining in the first panel (HP) looks rather unspecific. Could the authors use higher magnification pictures or co-staining with a plasma marker to convince the reader that the strong green signal in the lamina propria is real and not an artefact by the presence of high protein in these samples. Maybe flow cytometry of the lamina propria could be an option?
3. Figure 1g: The Ccl28 values seem to contain 2 outliers. Is there an explanation for these? If these were removed, would the data still be significantly increased in the HP group? This brings me to the next point:
4. Throughout the manuscript, I miss any information on how many times the individual experiments have been performed/repeated.
5. Figure 2: In order to claim that the phenotype of increased sIgA production is really independent of T cell help, I would like to see either data in TCRβd^{-/-} mice or in mice where T cells have been depleted with antibodies throughout the diet feeding. As the phenotype is T cell independent, most likely it already appears after less than 5 weeks of feeding, which would make the depletion experiment easier. Do the authors have data on how quick the phenotype appears? And is it reversible if you switch the mice back to a control diet?
6. Figure 3: While the data on Arip and Tgfb expression in the lamina propria are in line with the observed phenotype, Baff and Tslp expression seem to solely significantly decrease upon HF feeding which may be a completely different mechanism. Again 2 outliers in e. Are these the same mice as in Figure 1g? I think the text should be adjusted and not state a difference in Il10 expression.
7. Figure 4 and microbiota composition: The authors correctly conclude that feeding with different diets (HP, HC, HF) leads to intestinal microbial compositional shifts. For me, these shifts alone could well be responsible for the observed changes in sIgA levels. It would be nice that the authors include this possibility into their discussion or perform the necessary experiments to rule out this possibility. For example, feeding of monocolonized mice in which no microbial shift but an increase in EV production can be expected, could be one solution.
8. Figure 5a: The presented results of higher TLR4 activation in case of HP SI content are convincing. However, to prove the effect of EVs, the diets themselves should have also been tested for LPS contaminants. I cannot find any information on LPS contents in the three different diets. In case of differences, these may already lead to the difference in TLR4 activation. This control is absolutely required.
9. Figure 5: The contribution of TLR4 activation on intestinal epithelial cells to the sIgA increase in case of HP feeding should be tested in TLR4 KO mice. If possible in intestinal epithelial cell specific KO mice.
10. Figure 5: Use of the HT29 cell line: Certainly, a good model to start with, however, I would have wished to see the effect of EV numbers or intestinal content from differently fed mice directly on murine intestinal epithelial cells, for example by the use of intestinal organoids.
11. Would it be possible to treat GF mice with different numbers of EVs and see if these can induce IgA in the absence of intestinal microbiota? This would support the authors' view that the microbial shift observed in Figure 4 is not responsible for the difference in sIgA production.
12. Figure 6: Again, measurements of succinate, pyruvate and propionate in the diets themselves would have been an important control.
13. Figure 6: DSS is certainly an appropriate model to induce intestinal inflammation and ROS production by the microbiota. However, it seems difficult to me to use it as a model to induce more bacterial EVs. DSS itself leads to a dysbiosis which may be the cause for the change in the number of EVs/ml and not the increased stress on the microbiome. It has also been shown that DSS reduces EV production by certain members of the microbiota while it induces production by others (Kang et al., 2013, PMID: 24204633)

REVIEWER COMMENTS

Reviewer #1 (Remarks to the Author):

General comments:

Here, the authors propose a novel pathway through which dietary macronutrients influence sIgA production via gut microbiota-derived extracellular vesicles. This is an interesting paper that uses a range of techniques and tackles some exciting questions on the interaction between diet/microbiota/mucosal immunology. Unfortunately, however, the story isn't clear and some of the experiments conducted are not sufficient to explain the proposed (complex) pathway. The experiments conducted do not sufficiently support the claims that the effect on sIgA is driven exclusively by EVs and that EV production is driven by succinate. The major methodological concern relating to this is the isolation of cecal EVs and cecal SCFA rather than EVs or metabolites produced in the small intestine. This manuscript has the potential to be very good, however the proposed pathway has too many moving parts. It would benefit from a more focused approach on parts of this pathway to rigorously confirm their effect. Detailed comments are below, which if addressed, would make this manuscript suitable for publication.

Major comments:

1. As the focus of the manuscript is on small intestine, it is unclear why EVs and metabolites are isolated from the cecum and used for all following experiments. This is a major limitation that needs to be addressed.

We initially focused our study on cecum-derived EV as the cecum has the highest density of bacteria and thus higher number of bacteria-derived EV. Moreover, at this site, EV are predominantly derived from bacteria and not from other sources. However, we agree that the investigation of small intestine-derived EV is critical and have now repeated key experiments utilising microbiota EV isolated from the small intestine content.

As observed in the cecum, the concentration of EV in the small intestine is also higher under HP feeding conditions, with a peak diameter at approximately 175nm, which is within the expected size for bacterial EV (added to Fig. 5b):

As with cecum microbiota-derived EV, we have also confirmed that EV derived from the small intestine can also activate TLR4, and HP-derived EV activated TLR4 to a greater extent than EV from either HC- or HF-fed mice at physiological ratios (HP>HF=HC) (added to Fig. 5c):

HP small intestine microbiota-derived EV also significantly increased the expression of genes involved in T cell-independent IgA induction including *APRIL*, *CCL28* and *PIGR* compared to HC and HF derived EV, at physiological ratios (HP>HC=HF) (Fig. 5d-f):

Furthermore, these effects were via TLR4-dependent mechanisms as shown by the inhibition of downstream TLR4 signalling pathway using the antagonist BAY11-7082 (Fig. 5g):

These results are now in Fig. 5b-g, and the cecum-derived microbiota EV data presented initially in Figure 5 have been moved to Supplementary Fig. 4h-k.

We have now also quantified succinate levels in the small intestine content (now Fig. 6a), and as observed in the cecum, succinate was significantly increased in mice fed on a HP diet:

Overall, our results show that both small intestine and cecum-derived EV are increased under HP feeding conditions, along with succinate levels. Higher numbers of small intestine and cecum microbiota-EV from HP-fed mice similarly upregulated genes involved in T cell-independent sIgA production, through a TLR4-dependent pathway.

2. The cell line experiments appear crude and insufficient to support the story. It is not clear why HEK cells were used rather a small intestine cell line (e.g. HT-29). It is also not clear if this effect on gene expression in the cell line is purely due to a higher bacterial load in the assay. Other simple mouse experiments would be much more suited to investigate this (see some suggestions at the bottom)

The HEK cell line used in our study is a HEK-TLR4 reporter cell line, which allow us to determine whether TLR4 is activated and to what extent. We used this cell line to demonstrate that SI content as well as cecum-derived EV could activate TLR4 (Fig. 5a and Supplementary Fig. 4i). We have now confirmed that SI content-derived EV can also activate TLR4 (Fig. 5c).

We then demonstrated with the HT29 epithelial cell line that EV could directly upregulate the expression of genes involved in T cell-independent IgA pathways, and this was dependent on TLR signalling (Fig. 5d-g).

We have now clarified in the manuscript the choice of the cell line: **“To determine whether the small intestinal environment of HP-fed mice could differentially activate TLR4 signalling, we used the reporter cell line, HEK-Blue mTLR4, in which the intensity of TLR4 activation is assessed through a colorimetric assay.”**

We did not find changes in bacterial load in the small intestine, thus excluding this as an explanation for increased sIgA observed under HP feeding conditions (see answer to point 22).

3. The final DSS experiment does little to confirm the story relating to the story of protein, EVV, succinate etc.

We agree with the reviewer that the final DSS experiment was not sufficient to link our story. We have now restructured this section and added supporting data linking EV, succinate and DSS, as also suggested by reviewer 2.

IBD is associated with increased intestinal succinate levels (Lavelle and Sokol, 2020) and higher levels of sIgA (Lin et al., 2018). In the initial version of the manuscript, we show that DSS-induced colitis was also associated with higher levels of EV. We have now shown that under conditions of high succinate (either through HP feeding or direct administration of succinate in drinking water), mice develop more severe DSS colitis.

These data have been added to Fig. 6f-k and in the text (page 15): “Increased succinate has also been reported in human inflammatory bowel disease (IBD) as well as in DSS-induced colitis in mice^{27,28}. In both cases, increased TLR activation by microbial PAMPs has been reported in the gut²⁹. Accordingly, we found that mice treated with DSS had higher gut bacterial EV production with similar size distribution (Fig. 6d-e). As observed under HP feeding conditions, IBD is not only characterised by high gut succinate levels²⁷ but also increased TLR activation by gut microbial PAMPs²⁹. We found that DSS colitis was exacerbated under conditions of high succinate, either through HP feeding (Fig. 6f-h) or direct administration of succinate in drinking water (Fig. 6i-k). Our result suggests succinate is a key mediator that can contribute to an inflammatory intestinal environment by promoting the production of microbiota EV that can activate TLR4.”

Other comments:

1. The introduction (and the manuscript in general) is a little sparse on references. There are a number of other seminal manuscripts in the field that explore parts of this pathway with regards to sIgA production as influence by the gut microbiota etc.

We agree with the reviewer and have now added appropriate references throughout the introduction and discussion.

2. The results in figure 1e and a number of other figures appear to be driven by a small number of outliers. I would imagine that these are cage effects, which are commonly observed in microbiota studies. Please comment on whether this is the case, whether cage effects were adjusted for in any way, and whether potential cage effects effect the main results

Other than Fig. 1e, most variation in the data within group appears to be linked to inter-individual variability rather than cage effects. See graph in point 9 below.

We have also measured plasma IgA in additional cohorts and found no statistical differences across the diets, while similar trend towards less plasma IgA in HP-fed animals was observed:

3. Fig 1b is coloured by predicted sIgA concentrations as assumed by linear associations. It would be beneficial to present the R² of the model in order to indicate this. It would also be beneficial to the reader if the points representing the 10 diets were also enlarged. If possible, this panel should have scales matched to 1a in order for the graphs to be compared.

As suggested, we have now incorporated the R² value within both Fig. 1b and Fig. 1d and have enlarged the dots representing the 10 diets. We have also scaled Figure 1a to match the modelling presented in the manuscript. We thank the reviewer for these suggestions which have made the figures much easier to interpret for readers.

4. Could the authors comment somewhere in the manuscript as to whether total calorie/food intake differed between the 10 diet groups, particularly the HP/HC/HF groups?

Consistent with previous reports showing that food intake is regulated by protein consumption, known as the protein leverage hypothesis (Solon-Biet et al., 2014), mice fed on a HP diet had the lowest food intake, and thus lowest total energy intake (kJ/day) compared to those on lower protein diets (both HC and HF diet) (See figure below). This information has been added to the manuscript to Supplementary Fig. 1a-d, along with the existing data on macronutrient intake and sIgA levels.

5. In the sentence beginning with “For sIgA, model 1 had the most favourable AIC”, it is not clear to the reader what “model 1” is and how this would differ to model 2 or model 3 etc. If possible, rephrase to indicate e.g. the strongest model was driven by protein.

We have now rephrased the sentence, which provides better clarity to the readers on what the models represent (page 5): “Of the 4 models fitted to sIgA concentration (relating to first to fourth order of the Scheffé polynomials, described by Lawson and Wilden¹⁷), model 1 was the most appropriate as indicated by the lowest Akaike Information Criterion (AIC) value (Supplementary Table 2), which reveal that dietary protein concentration was the principal predictor of sIgA, there being a clearly graded increase in sIgA with increasing protein concentration, irrespective of fat or carbohydrate content ($R^2=0.8076$).”

6. It is not clear in the results how long the diets were fed for. Presumably all experiments were 5 weeks as stated in methods? Please state this somewhere in first paragraph of results.

We thank the reviewer for pointing out this omission. The dietary interventions were for 6 weeks, which has been included in the first paragraph of the results (page 5) and amended in the figure legends where appropriate: “To determine how dietary macronutrients might affect sIgA and thus host-microbiota mutualism, we fed mice on one of 10 isocaloric diets with defined ratios of macronutrients in the ranges 5-60% protein, 20-75% fat and 20-75% carbohydrate, for at least 6 weeks (Fig. 1a and Supplemental Table 1).”

7. Why are there only 4 data points in the HF diet in Fig 2b (compared with n=8 in HC?). There appears to be a relatively strong trend in the HF diet which may have been significant if this group also had 7-8 data points

We have repeated these experiments multiple times and we did not find consistent trends towards a decrease of B cells in MLN of HF fed animals when comparing the 3 diets:

8. In Fig 2c and 2e, the percentages in the flow cytometry plots don't seem to correspond with the corresponding bar plots in 2d and 2f.

We have cross-checked the data and could not find any inconsistencies between the percentages in the flow cytometry and the bar plots.

9. Again, in Fig 3c, there appears to be a cage effect probably? Please clarify if these results are driven by single cages or if they have genuine wide variation across all the animals in the group irrespective of cage

We find that in most cases these are genuine variation within groups, irrespective of cage. As illustrated below, similar outliers were observed even when the results were plotted by cage. The variation is thus not due to a cage effect but to interindividual variations.

10. Some of the terms used in the microbiota results are slightly confusing. E.g. how did you determine “adequate sequencing depth”. This could be approached in methods where you can detail that samples with less than X number of reads were removed.

We have now clarified this in the results section (page 9): “Furthermore, rarefaction analysis revealed adequate sequencing depth (total number of sequencing reads), with a horizontal asymptote evident for all samples (Supplementary Fig. 3c), demonstrating that our sequencing depth far exceeded what is required to uncover all observations (number of unique taxa, or ASV) present in each sample.”

Furthermore, we have also added more details to the figure legend of Supplementary Fig. 3c: “(c) Rarefaction analysis was performed for each sample which demonstrate that sequencing depth was sufficient to uncover all unique taxa (ASV) present in each sample.” We have also renamed the axis for increased clarity.

11. Similarly, higher diversity in a microbiome is very context specific. It is not clear if higher small intestine diversity is better, in fact it could be the opposite. It may be suitable to state that higher diversity in the stool microbiome is generally associated with health, however this should be saved for the discussion.

We agree that the existing statements regarding higher microbiome diversity had been generalised. We have now moved this from the results section into the discussion as suggested: “Our small intestinal microbiome data revealed no obvious dysbiosis associated with HP feeding that could explain the increase of sIgA. Rather, lower microbiota diversity was observed in HF-fed animals, an indicator of dysbiosis commonly observed in patients with allergies or IBD³⁰⁻³².” (page 16-17).

12. Again, the statement “suggesting a dysbiotic small intestinal microbiome” is misleading and should be removed from the results.

We have now removed this statement from the results.

13. Something that should be addressed in the discussion is the luminal vs adherent small intestinal microbiota. Emerging studies are beginning to investigate this and indicated that they differ quite significantly. This could have important implications for the context of this study with regards to contact with the epithelium, sIgA production etc.

We agree that this is a very interesting notion and we developed this in the discussion: “Additionally, phyla commonly associated with adherent intestinal bacteria, which are typically considered pro-inflammatory³³⁻³⁵, such as those from *Firmicutes*, *Bacteroidetes* and *Proteobacteria* did not differ between HP- and HC-fed groups.”, page 17.

14. If stating that HF had lower evenness in the results, you must explicitly state that it was not statistically significant

We have now reworded the results section stating that alpha diversity indices were statistically significant between groups: “However, HF-fed animals had lower diversity measures such as evenness and Inverse Simpson index, where Inverse Simpson index was significantly lower when compared to the HP group (Fig. 4a).” (page 9)

15. Please report PERMANOVA results (or equivalent) to show statistics that Unifrac distances were significantly different.

We have now included PERMANOVA analyses for both weighted and unweighted UniFrac distances showing that the small intestine microbiome composition is indeed significantly different between the different dietary groups. We have added these data to Supplementary Table 6.

We have now also included PCA plot of Aitchison distance – i.e. Euclidean distance of centre-log transformed count which is more appropriate for the analysis of compositional data, which is the case for microbiome data (Gloor et al., 2017). This, and the corresponding statistical analysis has been added to the main manuscript (Fig. 4b), while the PCoA plot has been moved to the supplementary (Supplementary Fig. 3d-e). We note that no difference in the conclusion using either analysis. This was added to the text (page 9-10): “By principal component analysis (PCA) of Aitchison distance, a compositionally appropriate between-sample distance metric, mice fed on the different diets had significantly distinct gut microbiota composition (Fig. 4b) as determined by PERMANOVA analysis (Supplementary Table 6). Likewise, principal coordinates analysis (PCoA) of UniFrac distances, a between-sample distance metric that considers phylogenetic relatedness, showed similar results, with HF feeding having the most distinct bacterial communities in both unweighted UniFrac PCoA and weighted UniFrac PCoA analyses (Supplemental Fig. 3d-e and Supplementary Table 6)”

PERMANOVA	HP vs HC	HP vs HF	HC vs HF
Unweighted UniFrac	R ² = 0.18861 P= 0.005	R ² = 0.34924 P= 2e-4	R ² = 0.35927 P= 3e-4
Weighted UniFrac	R ² = 0.45964 P= 2e-4	R ² = 0.67843 P= 2e-4	R ² = 0.81691 P= 2e-4
Aitchison’s distance	R ² = 0.35152 P= 3e-04	R ² = 0.53433 P= 1e-04	R ² = 0.52121 P= 3e-04

Furthermore, as requested by reviewer 2, we have also performed statistical analyses to identify all taxa that are differentially abundant. This data has been added to Supplementary Table 7-9.

16. It appears that there was a significant amount of cellulose in the high fat diet. This may have had a major effect on the gut microbiome in addition to the fat itself. Could the authors provide a line or two in the discussion about the limitations of the diets.

We agree this is an important point to discuss, and readers should be aware of qualitative aspect of the diets too. We have now included this in the discussion page 17: “Despite this, a HP microbiome had an exacerbated pro-inflammatory potential via increased production of microbiota EV that can activate TLR4, which was associated with exacerbated colitis compared to both HC- and HF-fed mice. Reduced colitis severity under HC feeding is likely due to increased levels of SCFA, which have been shown to be protective³⁶. HF had similarly reduced colitis either due to the macronutrient composition or to the high cellulose content used as filler in this diet. While cellulose is not a functional fibre and does not yield fermentable products such as SCFA, it has been shown to be protective against colitis³⁷. Future studies examining qualitative aspects of macronutrients, such as different types of fat and protein (i.e. animal vs. plant-based protein), would be required to fully appreciate the role of macronutrients on gut homeostasis as well as on the host sIgA responses. For example, soy-derived protein has been associated with lower sIgA, compared to casein-derived protein³⁸.”

17. Fig 4g, I would be surprised if there were only 15 taxa at genus level. Where are the others that make up the 100% relative abundance? The top 3 or 4 can't really be seen anyway so it may be best to present the top 10-12 and have an “others” section for all the other very low abundance taxa.

The reviewer is correct in that there are more than 15 taxa present at the genus level (total of 33 in our dataset), with the top 10 genera accounting for 98.1/94.2/94.6% of the total abundance for HP/HC/HF respectively and top 15 genera accounting for 98.8/97.6/98.6% of the total abundance for HP/HC/HF respectively.

In the previous version of the manuscript, we had only included the top 15 genera, which then makes up the 100% relative abundance. We agree that this is potentially misleading and thank the reviewer of this excellent suggestion. We have now changed the graph (now Fig. 4d) to include only the top 10 genera, with an additional ‘other’ category accounting for the other 23 genera.

18. This following statement: “While increased Proteobacteria have been linked to inflammation” is a bit of a generalization as proteobacteria is quite a broad phylum. Gammaproteobacteria are usually more inflammatory whilst other proteobacteria classes are less so. Again, I would save such statements until the discussion as they are context specific and can’t be generalized to the results.

We agree and have now removed this from the results section and incorporated it in the discussion instead.

19. Again, the statement “considered as probiotics” needs to be removed. Probiotics are defined species (see WHO definition). Avoid reference to probiotics and instead just refer to beneficial properties of these genera in the context of your study

We thank the reviewer for pointing out this oversight – we have now removed all inappropriate use of the word ‘probiotics’.

20. “By Spearman’s correlation, we confirmed that dietary protein content was more associated to sIgA than carbohydrate and fat (Fig. 4I).” This correlation is not shown in this heatmap.

We based this statement on the hierarchical clustering of the Spearman’s correlation (y-axis) – where pIgR was clustered more closely to protein content (P) than the other macronutrients. We believe it is not inaccurate to state that dietary protein was associated with higher sIgA levels based on this. We have removed this data as suggested below.

21. I am unsure if there is enough variation in the F, C and P contents of the three diets to conduct spearman correlations. Spearman are usually conducted on two continuous variables, however the F, C and P contents are not continuous outcomes. They each only have 2 values, either high (60/75%) or low (5/20%). This would be ok if you were looking across a broader scale of the 10 diets. The spearman correlations with sIgA/pIgR and microbial taxa are more suitable.

We agree and have now removed the correlation data.

22. What was cfu/g load in small intestine? Or at least 16S copy number per ug? The effect caused by HP microbiome may be based on higher bacterial load in the small intestine. This is important

We have quantified the bacterial load in the small intestine by comparison of DNA concentrations as well as by 16S copy number (both normalised to content weight). Based on these results, we found that there were no significant difference in bacterial load between the different dietary groups in the small intestine (Supplementary Fig. 4c-d):

Similarly, in the feces, HP-fed animals did not have the highest bacterial load.

Thus, the highest level of sIgA observed in HP-fed mice is not due to increased bacterial load in the small intestine, which is a very important addition to the manuscript. We have added the small intestine data to Supplementary Fig. 4c-d and added the following to the text (page 11):

“... as well as comparable bacterial load as determined by DNA concentration and total 16S copy number (Supplementary Fig. 4c-d).”

23. This approach seems a bit crude and a previous method should be cited. There are not enough details on this method e.g. concentration of small intestine inoculate that was applied to cells and the rationale?

We agree with the reviewer that the approach presented in Fig. 5a is crude, and this was intended to be. This was a pilot experiment, which used whole small intestine content as a proof-of-concept to show the greater capacity of the HP intestinal environment to activate TLR4. We later show that this increased capacity to activate TLR4 was due to EV (Fig. 5c). Furthermore, we have now detailed both in the text and figure legend the dilution and dosage used for these experiments.

24. Why HEK cells? Why not a small intestine cell line? Small intestine epithelium cell lines also express TLR4. HEK is also a human cell line. A mouse cell line would be more suited to a mouse microbiome. Much of this proposed pathway is also mucosally mediated. Cell lines can't replicate that. Please comment

Please see response to Major comment point 2 above regarding the choice of different cell lines used in our study.

We indeed used the epithelium cell line HT-29 in our study. HT29 is a colonic epithelial cell line derived from human colorectal adenocarcinoma. Due to its numerous similarities with small intestine enterocytes (Martínez-Maqueda et al., 2015), it has regularly been used by others in IgA related studies (Kitamura et al., 2000; Schneeman et al., 2005).

Using a human cell line provides a translational angle to the findings on top of mechanistic insights. Human vs mouse microbiomes are different while still share similarities with numerous studies showing transfer of human microbiota into mice (Koren et al., 2012; Ridaura et al., 2013) and the impact of mouse microbiota using HT29 was previously reported (Jahani-Sherafat et al., 2019; Macia et al., 2015). There is no good *in vitro* model that can replicate the complexity of the gut mucosal environment, we thus believe that combining *in vivo* and *in vitro* experiments is the best alternative with the current technology available.

25. Was cell count of inoculate or inoculate mass normalized beforehand in HEK experiment? Was it filtered in any way? More details/rationale needed on this experiment.

The inoculate used in Fig. 5a for the HEK experiments was normalised beforehand (diluted based on weight of content – 100mg/ml). This solution was then centrifuged to pellet and remove bacteria. This information is included in the methods, and we have added more details to the figure legend that precisely state the normalisation/dose being used.

26. TLR4 was not discussed before in the intro. This needs more rationale before introducing here. What was the expression of TLR4 in small intestines of mice fed the 3 diets?

We have now re-organised the introduction and better highlighted the pivotal role of TLR4 in the T cell-independent pathway of IgA induction, as well as highlighting LPS as the major ligand for TLR4 (page 3): “The majority of commensal bacteria induce T-cell independent IgA responses⁶, and sIgA generated via this pathway is of low affinity and polyreactive towards common bacterial antigens, limiting the attachment of a diverse range of bacteria to the host epithelium⁷. The induction of sIgA is regulated by the local cytokine environment, with the increase in class switch recombination (CSR) cytokines, APRIL and BAFF, produced by epithelial cells, promoting the differentiation of IgM⁺ B cells into IgA-producing plasma cells^{5,8}. These cytokines are produced as a result of toll-like receptor (TLR) activation, particularly TLR4, which recognises bacterial lipopolysaccharide (LPS)⁹.”

TLR activation by commensal bacteria is critical both for the recruitment of IgM⁺ B cells in the lamina propria, by increasing the expression of the gut homing chemokine CCL28⁹, and for the induction IgA CSR by upregulating APRIL¹⁰. TLR4 is the major TLR expressed on the epithelium and its overstimulation in TLR4 transgenic mice has been shown to promote CCL28 and APRIL production and accumulation of IgA in the lamina propria and gut lumen⁹.”

We thank the reviewer for this comment as changes in TLR4 expression might have accounted for the changes in the sIgA response between diets. Thus, we have quantified the expression level of TLR4 in the small intestine of mice fed on the 3 diets and found no differences across groups (see graph below).

This has now been added to manuscript (page 11): “Increased TLR4 signalling may be attributable to either higher expression of TLR4, increased concentration of its ligand, or increased exposure to its ligand through defects in the mucus layer or tight junction proteins. Mice fed on a HP, HC or HF diet had similar expression of TLR4 in the small intestine (Supplementary Fig. 4b), ...”

27. Please provide a reference for the EV isolation protocol. I am not familiar with EV isolation, however it appears to me that other extracellular products could be present after these centrifuging steps other than just EVs (e.g. ‘free’ metabolites).

We have now added a few references in the methods section for the EV isolation protocol. EV pellets are obtained after a final ultracentrifugation step (illustrated below), after removal of supernatant containing free metabolites. Differential centrifugation is considered the gold standard for EV isolation, particularly of bacterial-derived EV when sourced from the intestinal content, which is not contaminated with other host factors present in the blood such as low-density lipoprotein and other macromolecules.

Figure: Caecum or small intestinal content was homogenized in 0.02µm filtered PBS. Where appropriate, samples were first centrifuged at 340xg for 10min at 4°C to remove large debris. Supernatant was then centrifuged at

10,000xg for 20min at 4°C followed by 18,000xg for 45min at 4°C, filtered through a 0.22µm filter and centrifuged at 100,000xg for 2hr at 4°C. EV pellet was resuspended in 0.02µm filtered PBS.

28. Why cecum-derived EV? This is a critical concern. All of the work up to this point is focussed on the small intestine. Why not use SI-derived EVs?

As discussed above, we initially used cecum-derived EV due to its purity as well as the higher quantity of bacteria and thus bacteria-derived EV. We agree this is an important point and have repeated key experiments using microbiota EV derived from the small intestine. The data involving cecal-derived microbiota EV are now in the supplementary figures (Supplementary Fig. 4h-k) and SI-derived microbiota EV data in the main figures (Fig. 5b-g).

29. The effect of EVs on TLR4 was solely due to greater number of EVs. When the same doses were applied, there was no difference between diets. It is not clear if this was also the case for results presented in 5d-f. Were these normalized for EV quantity?

We have now clarified this important point in the figure legends.

30. Another limitation of this study is the lack of discussion about how/why EVs induce this response. What do they contain? It may be outside the scope, but needs to be addressed further in the discussion.

In our case, we found that the effect of EV on our phenotypes was due to the activation of TLR4 in a quantitative manner, which requires the physical interaction between EV and TLR4 expressed by the host. We agree that the content of EV is an important point too and have now discussed this in the Discussion (page 18): “Significant changes in the gut microbiota

composition have been shown to promote the production of different types of EV. **Indeed, EV can carry a range of molecules including proteins, metabolites and nucleic acids as cargo, and EV have been shown to mediate the virulence and pro-inflammatory effects of pathogenic bacteria²¹.**”

31. In Fig 5g, which diet are these EVs from? If from HP, it is not clear why this was conducted as there was not a clear effect of HP EVs on the expression of PIGR or APRIL compared with unstimulated controls or other diets

In the new experiments using small intestine-derived bacterial EV, EV derived from the HP group do significantly increase APRIL and PIGR. We show that this effect is also TLR4-dependent, and the group and exact experimental details has now been specified in the figure legends.

32. Why not isolate EVs from all three diets and gavage into different mice to then see effect on SI sIgA and TLR4/APRIL etc?

As demonstrated in Fig. 5c, the effect of EV on host sIgA response was due to the higher numbers of EV in the HP group and the activation of TLR4. Thus, administration of similar numbers of EV from animals fed on different diets will likely have no differential effects (Fig. 5c).

33. Were the three SCFA the only metabolites upregulated in HP or why were they chosen? More detail is needed on the NMR results and why the SCFA were chosen or if other metabolites were differentially abundant. Again, the major concern here is why NMR was conducted on cecal content rather than small intestine content. There are very few SCFA in the small intestine.

We focused our analysis on the major bacterial-derived metabolites detectable by NMR and that were increased in HP fed mice. We have now added more details to the methods (page 24-25): **“The Chenomx NMR Suite v8.4 software (Chenomx Inc) was used to identify and fit compounds from sample spectrum against pre-defined spectral reference libraries and concentrations quantified in reference to the internal standard.”**

We have now compared the metabolites found in different concentrations in HP-fed mice both in the small intestine and in the colon. Succinate was consistently increased in both sites under HP feeding conditions. As pyruvate and the SCFA propionate were not increased or not detectable in the small intestine, but only in the cecum, we hypothesise that they were not behind the high EV production in HP fed mice:

However, we found that branched chain amino acids (BCAA) were upregulated in both sites in HP-fed mice likely due to the high protein content of the diet. Microbial metabolism of amino acids such as BCAA could be involved in succinate production (Fischbach and Sonnenburg, 2011), which may explain why HP feeding is correlated with high succinate. While this is an interesting link, determining the exact pathway of succinate production is out of the scope of our study.

We have now provided a full list of metabolites detected in both the small intestine data (Supplementary Table 10) and the cecum (Supplementary Table 11). The new succinate data is now incorporated into Fig. 6a while the cecal data has been moved to Supplementary Fig. 5a.

34. “We found that, unlike EV-stimulated HT-29, neither pyruvate, propionate nor succinate could directly induce the expression of CCL28, APRIL or PIGR expression” - this statement should be revised as EVs did not stimulate APRIL and only HP EVs stimulated CCL28.

These data are now in the supplementary figures and have been replaced by the results involving the small intestine-derived EV in which HP-derived small intestine bacterial-EV increase the expression of APRIL, CCL28 and PIGR (Fig. 5d-f).

35. When comparing succinate with pyruvate and propionate for *E. coli* EV production, why has 10mM been used for succinate but 10-fold less conc used for the other SCFA (1mM)?

As mentioned above, pyruvate was not increased in the small intestine of HP-fed mice while the concentration of EV at this site was higher. Pyruvate is thus not an important factor regulating EV production. This has now been corrected in the manuscript.

36. Is the EVV result in Fig 6j normalized by microbiota cell count? Could it just be that DSS reduces cfu of small intestine microbiota

This is a very good point; it is conceivable that DSS could have an impact on the bacterial load. In human IBD as well as in DSS- or TNBS-induced colitis in mice the colonic microbial load has been reported to be increased (Hernández and Appleyard, 2003), through unknown mechanism. To exclude the effect of DSS on bacteria growth, we have tested the impact of 1 to 3% DSS supplemented media on *E. coli* growth and found no significant impact.

37. I am not sure how this last DSS experiment concludes the story? DSS alone could affect EV production and this does little to add to the EV/succinate story. This experiment either needs to be explored further to conclude the proposed pathway or removed entirely.

Please see answer to Major comment, point 3.

38. It would be beneficial to have a conceptual figure at the end that explains the proposed pathway. This may depend on editorial guidelines, however this complex pathway involving dietary protein, small intestine microbiota, EVs, sIgA, succinate, TLR4 etc etc is very hard to visualize and comprehend.

We agree and have now designed a graphical abstract (Fig. 7).

39. Further discussions is needed to on types of macronutrients. This study is limited by the chosen sources of macronutrients. Different sources could have highly varying effects (e.g. canola oil was use here as a fat source which is high in omega-6 fats but it is not representative of omega-3, MUFA, SFA etc. Similarly, carbohydrate sources have huge effect on microbiota whether simple sugars or complex carbohydrate structures).

The diets were designed based on AIN93G (Reeves et al., 1993), thus only the quantitative impact of macronutrients was the focus of this study. The sources of fat, protein and carbs used in AIN93G diet avoids deficiency of essential fatty acids and amino acids and keeps the diet palatable for the mice. However, we do agree that the qualitative aspect of macronutrients is very important as it has been shown that soy-derived protein was associated with lower sIgA compare to mice fed on diet with protein derived from casein (Zeng et al., 2020). It would indeed be interesting to study the impact of these diets on succinate and EV levels. The following sentence has been added in the discussion page 17: **“Future studies examining qualitative aspects of macronutrients, such as different types of fat and protein (i.e. animal vs. plant-based protein), would be required to fully appreciate the role of macronutrients on the host sIgA responses. For example, soy-derived protein has been associated with lower sIgA, compared to casein-derived protein³⁸.”**

40. Some alternative experiments that could help to clarify the story include:
- Treat mice (ideally germ-free or antibiotic treated) with small intestine contents of HP/HC/HF mice to demonstrate that the effect on sIgA/EVs is microbiota-dependent
 - Treat mice with isolated EVs from HP/HC/HF mice and measure TLR4/sIgA/APRIL etc to show more rigorous results *in vivo* rather than in HEK cells
 - Induce low-dose DSS colitis and treat mice with succinate or EVs to confirm it exacerbates the clinical effect.

These suggestions are excellent and discussed point-by-point below:

-Recolonisation experiments are usually done with cecal content or fecal transplantation. There is no evidence in the literature that the small intestine content would be appropriate.

-As discussed in point 32, the effect of EV on the host sIgA response was due to a quantitative rather than a qualitative effect. Thus, administration of the same numbers of EV from the different groups will likely not have a differential effect. However, we now provide evidence that TLR4 is a key pathway *in vivo* as *Tlr4*^{-/-} mice fed on HP diets do not have increase sIgA (see answer to point 9 to reviewer 3)

-We administered succinate in drinking water and found that these mice developed exacerbated DSS colitis (see below):

We have now added this data to Fig. 6i-k

Reviewer #2 (Remarks to the Author):

The manuscript entitled “Dietary protein increases T cell independent sIgA production through changes in gut microbiota-derived extracellular vesicles” by Tan et al. reports some interesting observation connecting diet-IgA production and microbiota. As much as the overall idea is intriguing there are some major limitation that reduce the enthusiasm for the manuscript.

General comments:

Authors claim “In our approach, we fed mice on one of 10 different isocaloric diets varying in their macronutrient composition and quantified sIgA. By using mixture modelling, we identified that dietary protein dramatically increased luminal sIgA levels, which is associated with increased expression of CCL28 and APRIL in the small intestine”. The majority of the manuscript data report only 3 diets “Animals were fed ad libitum on either a high protein, moderately low carbohydrate, moderately low-fat diet (P60, C20, F20; henceforth, HP); a low protein, high carbohydrate, moderately low-fat diet (P5, C75, F20; henceforth, HC), or a low protein, moderately low carbohydrate, high-fat diet (P5, C20, F75; henceforth, HF). These three diets represented the extremes (apices) from a larger set of 10 diet compositions (Supplementary Table 1) that were used in a subset of the experiments, encompassing a macronutrient range of protein (5-60%), carbohydrate (20-75%), and fat (20-75%) chosen using nutritional geometry to comprehensively sample dietary macronutrient mixture space”. It sounds like an overstatement to mention results obtained using 10 diets considering that these appear just in Figure 1a, b and d and Supplementary Figure 1. Only data obtained from 15-24 mice were showed. Indeed the total mice should be 80 (8 animals for each diet). Only data belonging to three diet (5-8 mice for diet) were reported. The low sample size does hardly allow generalizing those findings. Furthermore, analytical methods and statistics applied are not well described and hard to follow.

We agree that this sentence is misleading and have now reworded the following paragraph in the introduction (page 4): “In our approach, we fed mice on one of 10 different isocaloric diets varying in their macronutrient **composition, which identified an association between dietary protein and sIgA levels using mixture modelling. We validated and explored these results further using a subset of the 10 diets, representing a high-protein, high-carbohydrate or high-fat diet. We found that the high protein diet had the highest intestinal sIgA levels, and this was associated with increased expression of CCL28 and APRIL in the small intestine”**

To be clear, our modelling approach was to begin with a comprehensive coverage of macronutrient space using 10 diet formulations differing systematically in P, C and F and to use this analysis to define how macronutrient composition relates to host phenotypes such as host sIgA levels. This primary experiment and modelling led on to detailed studies using the 3 diets. These diets were chosen to represent the apices of the simplex triangle in macronutrient mixture space defined by the full array of 10 diets. These experiments were used to successfully validate our findings derived from the full 10 diets and to explore the results further. Hence, we confirmed that protein content drove sIgA and that this related to increased succinate level in the cecum (existing data) as well as in the small intestine, which drives the release of microbiota EV to promote TLR-dependent host sIgA induction (Fig. 5). These pathways were

conceived from the modelling data and validated with the 3 diets. With the exception of Fig. 2b (which we have validated), the remaining figures are from experiments with n=7-8 mice per group and validated by at least 2 independent experiments. We have now added this information to the figure legend of all the experiments, also as per the comments of reviewer #1.

We fully agree that the details of statistical analysis and analytical methods required to fully understand or replicate our study was lacking, and we have now addressed these issues (please see response to comments below).

The Methods section should contain all elements necessary for interpretation and replication of the results.

We have now added detailed description to the methods section necessary for interpretation and replication of results.

Please add references when appropriate.

We agree with the reviewer that the manuscript was a little sparse on references. We have now added appropriate references throughout the introduction and discussion.

Please add more details to permit the replication of the experiments.

As mentioned above, we have now added a detailed description to the methods section necessary for the interpretation and replication of results. We have further added additional references to the methods section, in particular for the EV isolation and microbiome analysis.

As mentioned, also by authors, IgA play a non redundant role in preventing microbes interaction with intestinal epithelial cells and possibly interdigitating dendritic cells. In this context it is crucial to evaluate the mucus layer thickness and integrity, furthermore, the role of the intestinal permeability should be considered (even with a simple qPCR of TJ genes).

We agree that this is an important point and that changes in gut homeostasis could potentially explain the differential effects of HP/HF/HC on sIgA levels.

We have now measured the mucus thickness in the small intestine of mice fed on the different diets and found no significant differences. This was added to Supplementary Fig. 4f. These new data confirm that the highest concentration of sIgA under HP feeding is not linked to a defect in the mucus barrier.

We have also quantified the expression of genes related to intestinal permeability by qPCR including *Tjp1* (ZO-1) and *Ocln* (Occludin) and found no differences between mice fed on HP/HF/HC. This has been added to Supplementary Fig. 4e, g:

We have reworded the manuscript accordingly (page 11): “**Increased TLR4 signalling may be attributed to either higher expression of TLR4, increased concentration of its ligand, or increased exposure to its ligand through defects in the mucus layer or tight junction proteins. Mice fed on a HP, HC or HF diet had similar expression of TLR4 in the small intestine (Supplementary Fig. 4b), as well as comparable bacterial loads as determined by DNA concentration and total 16S copy number (Supplementary Fig. 4c-d). We also quantified the levels of endotoxin in each diet and found low levels of endotoxin, which were comparable between the different diets (data not shown). Furthermore, the expression of gut barrier-related markers *Tjp1* (tight junction protein 1) and *Ocln* (occludin) were similar between groups (Supplementary Fig. 4e), as was the integrity of the mucus layer (Supplementary Fig. 4f) and *Muc2* (mucin 2) expression (Supplementary Fig. 4g).”**

Authors results encourage the idea that mice under different dietary regimes may better respond to acute inflammatory insult like DSS administration. Is this aspect been evaluated?

We agree that this is an interesting notion. IBD is also associated with increased intestinal succinate levels (Lavelle and Sokol, 2020) as well as increased TLR4 signalling by microbial PAMPs (Lu et al., 2018) similar to what we observe under HP feeding conditions. We have now assessed the severity of DSS colitis in mice fed on HP, HF or HC diets and found that mice fed on HP diet had the most severe clinical outcomes. This was added Fig. 6f-h:

Additional minor comments:

Numerous typographical and grammatical errors are littered through the manuscript and cause confusion over interpretation. For example L. 258: “metabolites that are significantly upregulated” Genes are upregulated; metabolites are not upregulated.

We thank the reviewer for pointing out this mistake and have replaced ‘upregulated’ with ‘increased’. We have now corrected the typographical and grammatical errors throughout the manuscript.

- L. 77-80: To determine how dietary macronutrients might affect sIgA and thus host-microbiota mutualism... Fig. 1a and Supplementary and Table 1 did not explicitly support the points being made by the authors.

The reference to Fig. 1a and Supplementary Table 1 was to highlight the composition of the diet rather than our approach of determining how macronutrient might affect sIgA. We have reworded the paragraph to make this clearer and have merged the corresponding paragraphs to improve the flow.

- L. 170-172. It is difficult to interpret in terms of its statistics, and does not appear to support the points being made by the authors (e.g., Supplementary Table 5).

We have now made it clearer in the graphs that there was no statistical difference when tested by ordinary one-way ANOVA, and this was the case for both post- and pre-processed reads. We have reworded the following line to clarify the link between Supplementary Table 5 and Supplementary Fig. 3a where the statistical analysis was presented (page 9): “Analysis demonstrated equal sequencing reads between samples (raw data presented in Supplementary Table 5 and corresponding graphs in Supplementary Fig. 3a-b) ...”

L. 173-179. How the diversity of the gut microbiota was estimated? The technical description in this paragraph appears jumbled. Descriptions of normalizations and other important parameters need to be included.

We thank the reviewer for these comments and agree that we have previously not provided enough details of our approach, which is essential for the interpretation of results and to ensure our results can be replicated by others.

We have now added a more detailed description of our analytical approach in the methods section (page 9), which include the description of the filtering, normalization and statistical methods used:

“Paired-end reads (2x250bp) were processed with the dada2 package (1.12.1) using R software (3.6.1) to generate amplicon sequence variant (ASV)⁴⁰. Taxonomic classification of ASV was performed using the Ribosomal Database Project (RDP) naive Bayesian classifier (rdp_train_set_16) with species level taxonomy (rdp_species_assignment_16) assignment (doi:10.5281/zenodo.801828). For downstream analysis, ASV with total abundance of less than 0.01% was first filtered out. For beta diversity analysis, a compositional data analysis approach was used⁴¹. ASV was centre log-ratio transformed after replacement of zeros via count zero multiplicative replacement⁴². Differences in microbiome composition between 2 groups was determined by PERMANOVA of Aitchison distance (Euclidean distance of CLR transformed data), after validation of homogeneity of group dispersions. For differential

abundance testing of ASV between 2 groups, ALDEx2 was used, and taxa were considered differentially abundant when Benjamini-Hochberg corrected expected p-value for both Welch's t test and Wilcoxon test were <0.05 and magnitude of effect size > 1. Analysis was performed with the following R packages: phyloseq (1.28.0), vegan (2.5.5), ALDEx2 (1.20.0), zCompositions (1.3.4), ranacapa (0.1.0). DNA sequencing data were deposited in the European Nucleotide Archive under accession number PRJEB39583."

Furthermore, we have also clarified the estimation of the microbiota diversity, which was done through multiple alpha diversity indices such as richness, evenness and Inverse Simpson index.

L. 180-197. Data were not supported by statistics. Please add the FDR values to support comparison among groups. The Authors do not show the data at the species level, why?

A limitation of conventional 16S rRNA gene sequencing is that it does not provide the resolution required to resolve at the species level (Johnson et al., 2019). For example, of the 110 ASV that are present in our dataset, only 4 were able to be assigned to a single species.

To support the comparison between groups, we have now performed differential abundance analysis using ALDEx2, a statistical method that is appropriate for compositional data. Statistical output (including FDR values) is now included in Supplementary table 7-9. The statistical output for Fig. 4e-g are now based on these FDR values.

L. 225: How EV was purified and characterized?

EV were purified by series of ultracentrifugation steps, as well as through filtration (see point 27 of reviewer 1). EV concentration and size distribution was analysed by nanoparticle tracking analysis (NTA). We have now added a detailed description of the methods used for the isolation and characterization of EV.

L. 253: Bacterial metabolites?

We have added more detail to the methods section in regard to quantification of bacterial metabolites by nuclear magnetic resonance (page 24-25): "For detection of polar metabolites by ¹H nuclear magnetic resonance spectroscopy (NMR), cecal or small intestinal content was homogenized in deuterium oxide (Sigma-Aldrich) at a concentration of 100mg/ml and centrifuged at 14,000xg for 5min at 4°C. For dietary metabolite analysis, diet pellets were suspended in 5ml deuterium oxide per gram diet and allowed to fully dissolve overnight at 4°C before clarification by centrifugation at 14,000xg for 5min at 4°C. Contents were then filtered through a 3kDa centrifugal filtration unit (Merck Millipore), and polar metabolites were collected from the aqueous phase of a chloroform-D/methanol-D/water extraction. Resulting solution was diluted with sodium triphosphate buffer (pH 7.0) containing 0.5mM 4,4-dimethyl-4-silapentane-1-sulfonic acid as internal standard (All from Sigma-Aldrich) and samples were run on a Bruker 600 MHz AVANCE III spectrometer. The Chenomx NMR Suite v8.4 software (Chenomx Inc) was used to identify and fit compounds from the acquired sample spectrum against pre-defined spectral reference libraries and concentrations quantified in reference to the internal standard."

Reviewer #3 (Remarks to the Author):

In this manuscript, Tan et al. report on the positive impact of high protein levels in diet on secretory IgA in the intestine. By screening mice fed for 5 weeks with 10 different purified diets that differed solely in their protein, carbohydrate and fat levels, but were comparable in all other micronutrients, in combination with mathematical modeling, high protein content in combination with low fat and low carbohydrate content was determined to be the optimal diet to increase sIgA levels in the intestine. The authors continue the manuscript by comparing a protein-rich (HP) diet with one enriched in carbohydrates (HC) and a high-fat diet (HF) and present experiments to show that the affected IgA production is T cell-independent and mucosa-specific. While the composition of the small intestinal microbiota is dramatically shifted upon feeding of the three diets, the authors claim that not the composition of the microbial community but rather their ability to release different amounts of extracellular vesicles is responsible for the increase in sIgA levels following HP feeding. By a combination of in vitro experiments with different cell lines, the authors show that HP-fed murine contents contained more EVs which are able to stimulate TLR4 on a coloncarcinoma cell line and induce IgA promoting proteins, such as APRIL and PIGR. The increased production of EVs under HP feeding seemed dependent on succinate-induced bacterial ROS production. While the overall topic of how dietary components can affect the intestinal host immune response either by themselves or through modification of the commensal microbiota is a research area of high importance to understand the host-microbial mutualism, and still under investigated, I am lacking important controls and additional experiments in this manuscript that are required to support the conclusions drawn by the authors. In my opinion, the manuscript is not acceptable for publication in Nature Communications in its present form and will need major revisions.

Please find my more detailed comments below:

1. The authors focus most of the paper on three diets with extreme protein, fat or carbohydrate content. While I can understand that testing all 10 diets *in vivo* is not feasible, I would have liked to see a control diet that is composed in regard to macronutrients as the regular mouse chow used in most animal facilities to have a baseline value for the read-outs. This would increase the physiological impact of the data as one could see which of the HP, HC and HF diet actually lies outside or around the “average” values in comparison to control diet.

Our study design, which utilises a quantitative modelling approach, that determines how specific components of macronutrients drive certain biological parameters. The diets essentially “hold hands” across a macronutrient response surface (Rodriguez Paris et al., 2020; Solon-Biet et al., 2014; Wali et al., 2021); it is not clear how a single outlying diet would feed in to inference from such an experiment. However, the composition of a diet representing chow can certainly be located within the macronutrient space covered by our 10 diets – it sits with a P:C:F ratio of ca. 20:60:20.

There are several issues in using regular chow as benchmark as the reviewer suggests. Even controlling for diet there can be substantial variation among institutions in animal phenotypes, thus a chow group would provide no external validation to our results. Second, chow as used in most animal facilities (including ours) is a cereal-based diet and its composition can vary substantially batch-to-batch due to the natural variability of the raw materials used. Thus, we based our diets on the semi-purified AIN93G control diet formulation which minimises batch-to-batch variability. When we compared our existing results to animals fed on the AIN93G control diet, we found that their levels of sIgA were comparable to those of mice fed on HC and HF, confirming the conclusion from the nutritional modelling based on 10 diets that high protein content induces robust host sIgA response:

2. Figure 1f: The IgA staining in the first panel (HP) looks rather unspecific. Could the authors use higher magnification pictures or co-staining with a plasma marker to convince the reader that the strong green signal in the lamina propria is real and not an artefact by the presence of high protein in these samples. Maybe flow cytometry of the lamina propria could be an option?

We have now confirmed these data by flow cytometry analysis on isolated small intestine lamina propria cells and we found significantly higher total numbers of plasma B cells

(identified as B220-IgA⁺) in HP-fed animals. This was confirmed by 2 independent experiments:

These data were added to the manuscript (page 6): “Consistent with this, mice fed on a HP diet had significantly greater number of B220-IgA⁺ plasma cells in the small intestinal lamina propria (Fig. 1g), as determined by flow cytometry (Gating strategy presented in Supplementary Fig. 1e).”

3. Figure 1g: The Ccl28 values seem to contain 2 outliers. Is there an explanation for these? If these were removed, would the data still be significantly increased in the HP group? This brings me to the next point:

This was due to an interindividual variability irrespective of a cage effect. As illustrated below, similar outliers were observed even when separated by cage:

Notwithstanding this variability, we consistently see significantly higher expression of key genes related to sIgA pathways in the HP group. Furthermore, the removal of the 2 outliers in the HP group did not change the statistical significance. Rather, it improved the statistical significance as shown below. However, we believe there is neither a technical nor a biological justification to exclude these samples.

4. Throughout the manuscript, I miss any information on how many times the individual experiments have been performed/repeated.

We thank the reviewer for pointing out this critical omission. We have now added the number of repeats for each experiment in the figure legends.

5. Figure 2: In order to claim that the phenotype of increased sIgA production is really independent of T cell help, I would like to see either data in TCRβ^{-/-} mice or in mice where T cells have been depleted with antibodies throughout the diet feeding.

We have now tested the impact of CD4⁺ T cell depletion on sIgA in HP-fed mice. Anti-CD4 antibody or isotype control were injected twice weekly intraperitoneally at a dose of 200μg per mice at the start of the HP feeding for 6 weeks. We confirmed by flow cytometry in lymphoid organs the efficacy of the treatment on CD4⁺ T cell depletion (added to Supplementary Fig. 2d):

We quantified sIgA in the small intestine by ELISA and found that mice fed on HP treated with isotype control or with anti-CD4 depleting antibodies had similar levels of sIgA, confirming that HP diet induced increase sIgA through a T cell independent mechanism:

This has been added to Supplementary Fig. 2c and the results (page 7): “To confirm this, we depleted CD4⁺ T cells in mice fed on HP diet from the start of the dietary intervention and found that mice treated with isotype or anti-CD4 depleting antibodies had comparable levels of sIgA (Supplementary Fig. 2c-d).”

As the phenotype is T cell independent, most likely it already appears after less than 5 weeks of feeding, which would make the depletion experiment easier. Do the authors have data on how quick the phenotype appears? And is it reversible if you switch the mice back to a control diet?

We have tracked faecal IgA levels as a proxy for intestinal IgA production to reduce the usage of animals. We found that 5 weeks on HP feeding was the minimum time necessary to induce consistent and significantly higher levels of sIgA (Supplementary Fig. 1f). While we also observed higher levels of sIgA in HP-fed mice after 1 week of feeding, this was not maintained.

To investigate whether the increase of sIgA induced by HP feeding was a dynamic and reversible process, we fed mice on HP for 6 weeks and then switched their diet towards HF for 6 weeks. Compared to mice maintained on HP, mice that were switched had a significant drop in their level of intestinal sIgA, demonstrating that the effects of HP on sIgA were reversible (Supplementary Fig. 1g):

This has been included in the manuscript (page 6-7): “Finally, we observed that a minimum of 5 weeks on HP feeding was necessary to stably increase sIgA levels (Supplementary Fig. 1f) and that this effect was a reversible process, as sIgA levels in HP-fed animals decreased when switched to a HF diet (Supplementary Fig. 1g). Together, these data show that protein is the major macronutrient driving sIgA production in the gut lumen and that this is reversible.”

6. Figure 3: While the data on April and Tgfb expression in the lamina propria are in line with the observed phenotype, Baff and Tslp expression seem to solely significantly decrease upon HF feeding which may be a completely different mechanism. Again 2 outliers in e. Are these the same mice as in Figure 1g? I think the text should be adjusted and not state a difference in Il10 expression.

We agree with the reviewer that different genes are likely to be regulated through different mechanisms, and that entire biological pathways are unlikely to be regulated by a single factor. Indeed, we focused our attention on those that are clearly driven by protein (such as CCL28, PIGR and APRIL, in our *in vitro* experiments in Figure 5). Removal of the 2 outliers in the HC group of Fig. 3e did not affect the statistics (it increased statistical significance vs. HF group):

These are indeed the same animals as in Fig. 1g, where no outliers were apparent for CCL28 expression. There is no technical or biological rationale for excluding these particular samples in Fig. 3e. We agree that the wording was misleading and have adjusted the text to precisely highlight whether *Tgfb* and *Il10* were statistically different or not (page 9): “We found that the expression of *Tgfb* was significantly higher in HP-fed mice and *Il10* was significantly lower in HF-fed mice compared to HC groups (Fig. 3e-f).”

7. Figure 4 and microbiota composition: The authors correctly conclude that feeding with different diets (HP, HC, HF) leads to intestinal microbial compositional shifts. For me, these shifts alone could well be responsible for the observed changes in sIgA levels. It would be nice that the authors include this possibility into their discussion or perform the necessary experiments to rule out this possibility. For example, feeding of monocolonized mice in which no microbial shift but an increase in EV production can be expected, could be one solution.

We agree that this shift itself could well be responsible for the effects on sIgA levels in HP-fed animals. However, it has been shown that the majority of host sIgA responses are derived from T cell-independent pathways initiated by small intestinal commensal bacteria without specificity to any particular species, with few exceptions such as SFB and *Mucispirillum* which induces host sIgA specifically through T cell-dependent pathways (Bunker et al., 2015). We have now discussed this and have highlighted that we did not detect these SFB and *Mucispirillum* in our animals: “Of note, we did not identify segmented filamentous bacteria or *Mucispirillum* in our animals (Supplementary Table 7-9), which are two genera known to induce strong T cell-dependent sIgA responses⁶.” (page 10). Furthermore, we did not see any changes to T cell-dependent pathways (Fig. 2), suggesting it is unlikely that changes to specific taxa are driving differences in sIgA response.

Additionally, we have added new data demonstrating that direct administration of purified EV to mice *in vivo* lead to significant increase of sIgA levels (Fig. 5j), confirming the effect of EV on sIgA.

8. Figure 5a: The presented results of higher TLR4 activation in case of HP SI content are convincing. However, to prove the effect of EVs, the diets themselves should have also been tested for LPS contaminants. I cannot find any information on LPS contents in the three different diets. In case of differences, these may already lead to the difference in TLR4 activation. This control is absolutely required.

We agree this is an important control – and have now tested endotoxin levels from 2 different batches of each diet (HP/HC/HF). The averaged endotoxin levels are summarised below, and a note of this was added to the manuscript in the methods (page 20) “... and the HP/HC/HF used for most experiments had very low endotoxin levels as tested with the PyroGene Recombinant Factor C kit (Lonza)” and in the results section (page 11): “We also quantified the levels of endotoxin in each diet and found low levels of endotoxin, which were comparable between the different diets (data not shown).”

	HP	HC	HF
ENDOTOXIN (EU/ML)	0.013691	0.053397	ND

We note that endotoxin levels are very low in all of the diets tested (and was not at detectable levels in the HF diets), much lower than the 0.5EU/ml threshold typically require for medical devices. Furthermore, HC diets had approximately 4X higher endotoxin than HP diets. This excludes the possibility that higher sIgA in HP-fed mice may be due to the contamination of the diet.

9. Figure 5: The contribution of TLR4 activation on intestinal epithelial cells to the sIgA increase in case of HP feeding should be tested in TLR4 KO mice. If possible in intestinal epithelial cell specific KO mice.

We have now tested the effect of HP diet feeding in TLR4 KO mice. When compared to wildtype animals fed on a HP diet, sIgA levels in TLR4 KO mice were almost 2-fold lower than wild-type animals. This result confirms our *in vitro* finding that TLR4 is an important pathway involved in the impact of HP-EV on sIgA. To exclude that TLR4KO mice had less sIgA potentially due to less bacterial EV, we quantified EV in the small intestine of these animals. On the contrary, TLR4KO mice had the highest levels of EV, which confirmed that a proper EV-TLR4 signalling is necessary for sIgA induction.

We have added the sIgA data to the manuscript (Supplementary Fig. 4a): “TLR4 is one of the main TLRs expressed by the small intestinal enterocytes and TLR4 signalling is the major pathway promoting T cell-independent IgA responses⁹. To determine whether TLR4 signalling was linked to the effect of HP *in vivo*, we fed wild type (WT) versus *Tlr4*^{-/-} mice on HP diets for 6 weeks and measured sIgA levels in the small intestine. The absence of TLR4 abrogated the effects of HP on sIgA levels (Supplementary Fig. 4a), confirming that a HP diet promote sIgA production via TLR4-dependent mechanisms.”

10. Figure 5: Use of the HT29 cell line: Certainly, a good model to start with, however, I would have wished to see the effect of EV numbers or intestinal content from differently fed mice directly on murine intestinal epithelial cells, for example by the use of intestinal organoids.

HT29 is a colonic epithelial cell line derived from human colorectal adenocarcinoma. Due to its numerous similarities with small intestine enterocytes (Martínez-Maqueda et al., 2015), it has frequently been used by others in IgA-related studies (Kitamura et al., 2000; Schneeman et al., 2005). We believe that HT29 is a valid model in our study, however, we agree that it is important to see the effect of EV in different models.

As HT29 is a good approach but fails to mimic the mucosal environment, we demonstrate that *in vivo* oral administration of EV to mice lead to increased sIgA. This *in vivo* approach is more physiological than organoids and is a good complement to our previous findings. These data were added to the manuscript in figure Fig. 5j and further discussed in point 7 above.

11. Would it be possible to treat GF mice with different numbers of EVs and see if these can induce IgA in the absence of intestinal microbiota? This would support the authors' view that the microbial shift observed in Figure 4 is not responsible for the difference in sIgA production.

Frequent gavaging of GF mice is a very challenging approach with high risks of contamination. Moreover, these mice have numerous physiological defects, thus any findings would need to be validated in more physiological models. Instead, as mentioned in point 7, we have orally administered daily EV to SPF mice and found that it significantly increased sIgA, compared to mice given vehicle control (PBS) (Fig. 5j).

12. Figure 6: Again, measurements of succinate, pyruvate and proprionate in the diets themselves would have been an important control.

As these diets are semi-purified (i.e. purified ingredients such as amino acids are used to manufacture the diet), we do not expect levels of these bacterial-derived fermentation products to be present. Nevertheless, we have confirmed this by NMR analysis and no levels of succinate, pyruvate nor propionate were found. See example below with succinate. We have noted this in the manuscript (page 14): “Of note, succinate was not detectable in the diets used in this study (data not shown).”

13. Figure 6: DSS is certainly an appropriate model to induce intestinal inflammation and ROS production by the microbiota. However, it seems difficult to me to use it as a model to induce more bacterial EVs. DSS itself leads to a dysbiosis which may be the cause for the change in the number of EVs/ml and not the increased stress on the microbiome. It has also been shown that DSS reduces EV production by certain members of the microbiota while it induces production by others (Kang et al., 2013, PMID: 24204633)

We agree that DSS can have a direct and confounding effect in our experiment. We have now tested whether colitis severity was different in mice fed on HC, HF or HP diet. HP-fed mice developed significantly more severe colitis than mice fed on HC or HF diet. This additional experiment fits nicely with the fact that IBD has been linked to higher sIgA levels (Lin et al., 2018), which we observed in the HP-fed mice. We thus hypothesise that the high EV released under HP feeding conditions can prime for IBD through TLR4 activation.

References

- Bunker, J.J., Flynn, T.M., Koval, J.C., Shaw, D.G., Meisel, M., McDonald, B.D., Ishizuka, I.E., Dent, A.L., Wilson, P.C., Jabri, B., Antonopoulos, D.A., Bendelac, A., 2015. Innate and Adaptive Humoral Responses Coat Distinct Commensal Bacteria with Immunoglobulin A. *Immunity* 43, 541–553. <https://doi.org/10.1016/j.immuni.2015.08.007>
- Fischbach, M.A., Sonnenburg, J.L., 2011. Eating for two: how metabolism establishes interspecies interactions in the gut. *Cell Host Microbe* 10, 336–347. <https://doi.org/10.1016/j.chom.2011.10.002>
- Gloor, G.B., Macklaim, J.M., Pawlowsky-Glahn, V., Egozcue, J.J., 2017. Microbiome Datasets Are Compositional: And This Is Not Optional. *Front. Microbiol.* 8, 2224. <https://doi.org/10.3389/fmicb.2017.02224>
- He, B., Xu, W., Santini, P.A., Polydorides, A.D., Chiu, A., Estrella, J., Shan, M., Chadburn, A., Villanacci, V., Plebani, A., Knowles, D.M., Rescigno, M., Cerutti, A., 2007. Intestinal bacteria trigger T cell-independent immunoglobulin A(2) class switching by inducing epithelial-cell secretion of the cytokine APRIL. *Immunity* 26, 812–826. <https://doi.org/10.1016/j.immuni.2007.04.014>
- Hernández, G.A., Appleyard, C.B., 2003. Bacterial load in animal models of acute and chronic “reactivated” colitis. *Digestion* 67, 161–169. <https://doi.org/10.1159/000071296>
- Jahani-Sherafat, S., Azimirad, M., Ghasemian-Safaei, H., Ahmadi Amoli, H., Moghim, S., Sherkat, G., Zali, M.R., 2019. The effect of intestinal microbiota metabolites on HT29 cell line using MTT method in patients with colorectal cancer. *Gastroenterol. Hepatol. Bed Bench* 12, S74–S79.
- Johnson, J.S., Spakowicz, D.J., Hong, B.-Y., Petersen, L.M., Demkowicz, P., Chen, L., Leopold, S.R., Hanson, B.M., Agresta, H.O., Gerstein, M., Sodergren, E., Weinstock, G.M., 2019. Evaluation of 16S rRNA gene sequencing for species and strain-level microbiome analysis. *Nat. Commun.* 10, 5029. <https://doi.org/10.1038/s41467-019-13036-1>
- Kim, Y., Hwang, S.W., Kim, S., Lee, Y.-S., Kim, T.-Y., Lee, S.-H., Kim, S.J., Yoo, H.J., Kim, E.N., Kweon, M.-N., 2020. Dietary cellulose prevents gut inflammation by modulating lipid metabolism and gut microbiota. *Gut Microbes* 11, 944–961. <https://doi.org/10.1080/19490976.2020.1730149>
- Kitamura, T., Garofalo, R.P., Kamijo, A., Hammond, D.K., Oka, J.A., Caflisch, C.R., Shenoy, M., Casola, A., Weigel, P.H., Goldblum, R.M., 2000. Human Intestinal Epithelial Cells Express a Novel Receptor for IgA. *J. Immunol.* 164, 5029–5034. <https://doi.org/10.4049/jimmunol.164.10.5029>
- Koren, O., Goodrich, J.K., Cullender, T.C., Spor, A., Laitinen, K., Bäckhed, H.K., Gonzalez, A., Werner, J.J., Angenent, L.T., Knight, R., Bäckhed, F., Isolauri, E., Salminen, S., Ley, R.E., 2012. Host remodeling of the gut microbiome and metabolic changes during pregnancy. *Cell* 150, 470–480. <https://doi.org/10.1016/j.cell.2012.07.008>
- Lavelle, A., Sokol, H., 2020. Gut microbiota-derived metabolites as key actors in inflammatory bowel disease. *Nat. Rev. Gastroenterol. Hepatol.* 17, 223–237. <https://doi.org/10.1038/s41575-019-0258-z>
- Lin, R., Chen, H., Shu, W., Sun, M., Fang, L., Shi, Y., Pang, Z., Wu, W., Liu, Z., 2018. Clinical significance of soluble immunoglobulins A and G and their coated bacteria in feces of patients with inflammatory bowel disease. *J. Transl. Med.* 16, 359. <https://doi.org/10.1186/s12967-018-1723-0>

- Lu, Y., Li, X., Liu, S., Zhang, Y., Zhang, D., 2018. Toll-like Receptors and Inflammatory Bowel Disease. *Front. Immunol.* 9, 72. <https://doi.org/10.3389/fimmu.2018.00072>
- Macia, L., Tan, J., Vieira, A.T., Leach, K., Stanley, D., Luong, S., Maruya, M., Ian McKenzie, C., Hijikata, A., Wong, C., Binge, L., Thorburn, A.N., Chevalier, N., Ang, C., Marino, E., Robert, R., Offermanns, S., Teixeira, M.M., Moore, R.J., Flavell, R.A., Fagarasan, S., Mackay, C.R., 2015. Metabolite-sensing receptors GPR43 and GPR109A facilitate dietary fibre-induced gut homeostasis through regulation of the inflammasome. *Nat. Commun.* 6, 6734. <https://doi.org/10.1038/ncomms7734>
- Martínez-Maqueda, D., Miralles, B., Recio, I., 2015. HT29 Cell Line, in: Verhoeckx, K., Cotter, P., López-Expósito, I., Kleiveland, C., Lea, T., Mackie, A., Requena, T., Swiatecka, D., Wichers, H. (Eds.), *The Impact of Food Bioactives on Health: In Vitro and Ex Vivo Models*. Springer, Cham (CH).
- Reeves, P.G., Nielsen, F.H., Fahey, G.C., 1993. AIN-93 purified diets for laboratory rodents: final report of the American Institute of Nutrition ad hoc writing committee on the reformulation of the AIN-76A rodent diet. *J. Nutr.* 123, 1939–1951. <https://doi.org/10.1093/jn/123.11.1939>
- Ridaura, V.K., Faith, J.J., Rey, F.E., Cheng, J., Duncan, A.E., Kau, A.L., Griffin, N.W., Lombard, V., Henrissat, B., Bain, J.R., Muehlbauer, M.J., Ilkayeva, O., Semenkovich, C.F., Funai, K., Hayashi, D.K., Lyle, B.J., Martini, M.C., Ursell, L.K., Clemente, J.C., Van Treuren, W., Walters, W.A., Knight, R., Newgard, C.B., Heath, A.C., Gordon, J.I., 2013. Gut microbiota from twins discordant for obesity modulate metabolism in mice. *Science* 341, 1241214. <https://doi.org/10.1126/science.1241214>
- Rodriguez Paris, V., Solon-Biet, S.M., Senior, A.M., Edwards, M.C., Desai, R., Tedla, N., Cox, M.J., Ledger, W.L., Gilchrist, R.B., Simpson, S.J., Handelsman, D.J., Walters, K.A., 2020. Defining the impact of dietary macronutrient balance on PCOS traits. *Nat. Commun.* 11, 5262. <https://doi.org/10.1038/s41467-020-19003-5>
- Schneeman, T.A., Bruno, M.E.C., Schjerven, H., Johansen, F.-E., Chady, L., Kaetzel, C.S., 2005. Regulation of the polymeric Ig receptor by signaling through TLRs 3 and 4: linking innate and adaptive immune responses. *J. Immunol. Baltim. Md 1950* 175, 376–384. <https://doi.org/10.4049/jimmunol.175.1.376>
- Solon-Biet, S.M., McMahon, A.C., Ballard, J.W.O., Ruohonen, K., Wu, L.E., Cogger, V.C., Warren, A., Huang, X., Pichaud, N., Melvin, R.G., Gokarn, R., Khalil, M., Turner, N., Cooney, G.J., Sinclair, D.A., Raubenheimer, D., Le Couteur, D.G., Simpson, S.J., 2014. The ratio of macronutrients, not caloric intake, dictates cardiometabolic health, aging, and longevity in ad libitum-fed mice. *Cell Metab.* 19, 418–430. <https://doi.org/10.1016/j.cmet.2014.02.009>
- Wali, J.A., Milner, A.J., Luk, A.W.S., Pulpitel, T.J., Dodgson, T., Facey, H.J.W., Wahl, D., Kebede, M.A., Senior, A.M., Sullivan, M.A., Brandon, A.E., Yau, B., Lockwood, G.P., Koay, Y.C., Ribeiro, R., Solon-Biet, S.M., Bell-Anderson, K.S., O’Sullivan, J.F., Macia, L., Forbes, J.M., Cooney, G.J., Cogger, V.C., Holmes, A., Raubenheimer, D., Le Couteur, D.G., Simpson, S.J., 2021. Impact of dietary carbohydrate type and protein-carbohydrate interaction on metabolic health. *Nat. Metab.* 3, 810–828. <https://doi.org/10.1038/s42255-021-00393-9>
- Zakrzewski, M., Proietti, C., Ellis, J.J., Hasan, S., Brion, M.-J., Berger, B., Krause, L., 2017. Calypso: a user-friendly web-server for mining and visualizing microbiome–environment interactions. *Bioinformatics* 33, 782–783. <https://doi.org/10.1093/bioinformatics/btw725>
- Zeng, B., Wang, D., Wang, H., Chen, T., Luo, J., Xi, Q., Sun, J., Zhang, Y., 2020. Dietary Soy Protein Isolate Attenuates Intestinal Immunoglobulin and Mucin Expression in

Young Mice Compared with Casein. *Nutrients* 12, 2739.
<https://doi.org/10.3390/nu12092739>

REVIEWER COMMENTS

Reviewer #1 (Remarks to the Author):

Here, the authors present a revised version of their manuscript that details a novel link between dietary protein, sIgA and microbiota-derived extracellular vesicles. Additional animal experiments have been added to the original manuscript to address concerns about small-intestine derived EVs, the effect of succinate on DSS induced colitis and other concerns which has hugely benefitted the manuscript.

I am extremely impressed by the lengths the authors have gone to to address every point by the 3 reviewers. It appears that extensive work has been conducted through new animal experiments and other assays to address every scientific query posed by the reviewers (and all in the midst of a global pandemic!). These additional experiments and revisions are more than sufficient to conclude this story and have confirmed the proposed pathway.

This is an excellent paper and I would like to commend the authors for their efforts!

Ruairi Robertson

Reviewer #3 (Remarks to the Author):

Overall, the revised manuscript has largely improved compared to the first version and is now in a very good state. The authors have also responded to all my comments and I am largely pleased with the answers. There remain only two points where in my opinion the newly presented experiments miss again an important control:

A. Previous comment 5: As suggested by myself, the reviewers have performed the CD4 T cell depletion experiment on mice fed HO diet and clearly see no involvement of CD4 T cells in the measured sIgA levels. However, to rule out that the antibody treatment itself does not already alter the sIgA levels, I would have liked to see the experiment performed in parallel in mice fed a control diet with lower IgA levels.

B. Previous comment 9: The experiment on Tlr4^{-/-} mice is a nice addition to the manuscript. Again, I would have liked to see the HP AND HF feeding on Tlr4^{-/-} mice to see that HF on Tlr4^{-/-} do not have even lower sIgA levels compared to Tlr4^{-/-} HP fed mice. It is known that TLR4 generally contributes to IgA production as also now indicated in the manuscript and I would have liked to see that HP and HF fed Tlr4^{-/-} mice have the same low level of IgA.

REVIEWER COMMENTS (Round 2)

Reviewer #3 (Remarks to the Author):

Overall, the revised manuscript has largely improved compared to the first version and is now in a very good state. The authors have also responded to all my comments and I am largely pleased with the answers. There remain only two points where in my opinion the newly presented experiments miss again an important control:

A. Previous comment 5: As suggested by myself, the reviewers have performed the CD4 T cell depletion experiment on mice fed HO diet and clearly see no involvement of CD4 T cells in the measured sIgA levels. However, to rule out that the antibody treatment itself does not already alter the sIgA levels, I would have liked to see the experiment performed in parallel in mice fed a control diet with lower IgA levels.

To establish that the antibody treatment by itself did not affect sIgA level in mice, we have now repeated the experiment in mice fed on a high protein diet (HP) or an AIN93G control diet (CT) as suggested (Graph below). We have confirmed that HP fed mice had higher sIgA levels than CT fed mice under control conditions in mice treated with the isotype control (CT iso vs HP iso) and under CD4 T cell depletion condition (CT aCD4 and HP aCD4). We also confirmed that depletion of CD4 T cells did not affect sIgA either in the controls or in the HP fed mice, ensuring that the impact of HP feeding on sIgA level is mediated through a T cell-independent mechanism. These new data are now in Supplementary Figure 2c, replacing the previous data in which only HP fed mice were presented.

The following were also amended in the manuscript: “To confirm this, we depleted CD4⁺ T cells in mice fed on HP diet from the start of the dietary intervention and found that mice treated with isotype or anti-CD4 depleting antibodies had comparable levels of sIgA (Supplementary Fig. 2c-d). **Mice fed on control diet had lower sIgA than HP fed mice regardless of the depletion of CD4 T cells.** Together, our results show that HP feeding promotes high levels of small intestine sIgA via a T cell-independent pathway.” On pages 7-8.

B. Previous comment 9: The experiment on Tlr4^{-/-} mice is a nice addition to the manuscript. Again, I would have liked to see the HP AND HF feeding on Tlr4^{-/-} mice to see that HF on Tlr4^{-/-} do not have even lower sIgA levels compared to Tlr4^{-/-} HP fed mice. It is known that TLR4 generally contributes to IgA production as also now indicated in the manuscript and I would have liked to see that HP and HF fed Tlr4^{-/-} mice have the same low level of IgA.

We agree that *Tlr4*^{-/-} mice were a powerful tool to show that a high protein diet affected sIgA via TLR4. We believe that the current data adequately addresses whether (1) HP diet increases sIgA levels and (2) whether this effect is dependent on TLR4 signalling.

Investigating the impact of HF feeding is a very interesting suggestion relevant for projects in the future. However, we respectfully submit that this is beyond the scope of this manuscript, which is to delineate the mechanism by which HP diet increases sIgA levels.

Reviewer #1 (Remarks to the Author):

Here, the authors present a revised version of their manuscript that details a novel link between dietary protein, sIGA and microbiota-derived extracellular vesicles. Additional animal experiments have been added to the original manuscript to address concerns about small-intestine derived EVs, the effect of succinate on DSS induced colitis and other concerns which has hugely benefitted the manuscript.

I am extremely impressed by the lengths the authors have gone to to address every point by the 3 reviewers. It appears that extensive work has been conducted through new animal experiments and other assays to address every scientific query posed by the reviewers (and all in the midst of a global pandemic!). These additional experiments and revisions are more than sufficient to conclude this story and have confirmed the proposed pathway.

This is an excellent paper and I would like to commend the authors for their efforts!

Ruairi Robertson